# Subcellular sorting of neuregulins controls the assembly of excitatory-inhibitory cortical circuits

**David Exposito-Alonso[1,2†], Catarina Osório[1,2†], Clémence Bernard[1,2], Sandra Pascual-García[1], Isabel del Pino[1], Oscar Marín[1,2*], Beatriz Rico[1,2*]**

[1]Centre for Developmental Neurobiology, Institute of Psychiatry, Psychology and Neuroscience, King's College London, London, United Kingdom; [2]MRC Centre for Neurodevelopmental Disorders, King's College London, London, United Kingdom

**Abstract** The assembly of specific neuronal circuits relies on the expression of complementary molecular programs in presynaptic and postsynaptic neurons. In the cerebral cortex, the tyrosine kinase receptor ErbB4 is critical for the wiring of specific populations of GABAergic interneurons, in which it paradoxically regulates both the formation of inhibitory synapses as well as the development of excitatory synapses received by these cells. Here, we found that Nrg1 and Nrg3, two members of the neuregulin family of trophic factors, regulate the inhibitory outputs and excitatory inputs of interneurons in the mouse cerebral cortex, respectively. The differential role of Nrg1 and Nrg3 in this process is not due to their receptor-binding EGF-like domain, but rather to their distinctive subcellular localization within pyramidal cells. Our study reveals a novel strategy for the assembly of cortical circuits that involves the differential subcellular sorting of family-related synaptic proteins.

**\*For correspondence:**
oscar.marin@kcl.ac.uk (OM);
beatriz.rico@kcl.ac.uk (BR)

[†]These authors contributed equally to this work

**Competing interests:** The authors declare that no competing interests exist.

## Introduction

The function of the cerebral cortex relies on dynamic interactions between excitatory and inhibitory neurons. These interactions are governed by precise synaptic connectivity patterns, which are laid down during development and subsequently sculpted by ongoing activity to support neural processes such as learning. What are the molecular and cellular rules that underlie the development of specific connectivity motifs? Neurons use cellular and subcellular strategies for the assembly of specific synapses (*de Wit and Ghosh, 2016*; *Südhof, 2018*; *Yogev and Shen, 2014*). Cellular specificity in synaptic targeting involves the generation of multiple protein isoforms of the same gene family through alternative splicing (*de Wit and Ghosh, 2016*). For instance, the expression of a random combinatory pool of protocadherin isoforms provides a cell-surface recognition code that is mediated by homophilic interactions (*Thu et al., 2014*). The alternative splicing of neurexins, which is regulated in a cell-specific manner, also supports synaptic specificity at the cellular level (*Fuccillo et al., 2015*; *Iijima et al., 2011*). Interestingly, the establishment of specific patterns of subcellular connectivity relies on the expression of complementary molecular programs in presynaptic and postsynaptic neurons, which provide unique address codes for the subcellular targeting of connections (*Apóstolo and de Wit, 2019*; *Favuzzi and Rico, 2018*). For example, expression of latrophilins at specific postsynaptic sites drives the assembly of excitatory inputs into different dendritic compartments in pyramidal cells (*Sando et al., 2019*). Similarly, expression of Cbln4 in a subclass of cortical interneurons is required for the targeting of inhibitory inputs to the dendrites of pyramidal cells (*Favuzzi et al., 2019*).

The molecular mechanisms underlying subcellular synaptic specificity likely rely on the sorting of specific mRNAs and/or proteins into distinct subcellular compartments. For instance, the localization

of certain mRNAs into the somatodendritic or axonal compartments of neurons depends on alternative 3'UTR sequences (*Cioni et al., 2019*; *Tushev et al., 2018*). In addition, it has been shown that synaptic proteins localize to different subcellular compartments by either selective retention or selective delivery (*Ribeiro et al., 2018*; *Sampo et al., 2003*). These processes are thought to be mediated by the compartmentalized organization of the cytoskeleton and specific interactions with sorting receptors, adaptors, and motor proteins (*Gumy and Hoogenraad, 2018*; *Ribeiro et al., 2019*).

The tyrosine kinase receptor ErbB4 is involved in the wiring of specific microcircuits in the cerebral cortex (*Del Pino et al., 2018*). ErbB4 is uniquely expressed by some types of cortical interneurons and absent from pyramidal cells (*Fazzari et al., 2010*; *Vullhorst et al., 2009*). ErbB4 localizes to synaptic contacts in both the somatodendritic (postsynaptic) and/or axonal (presynaptic) compartments of interneurons (*Fazzari et al., 2010*), where it regulates both the number of excitatory synapses received and inhibitory synapses made by different types of interneurons (*Del Pino et al., 2017*; *Del Pino et al., 2013*; *Fazzari et al., 2010*; *Ting et al., 2011*; *Yang et al., 2013*). These observations raise the possibility that presynaptic and postsynaptic ErbB4 receptors may interact with different synaptic partners during the development of cortical circuits.

Neuregulins comprise a family of growth factors that activate receptor tyrosine kinases of the ErbB family (*Mei and Nave, 2014*). Type III Neuregulin 1 (hereafter referred to as Nrg1) and Neuregulin 3 (Nrg3) are the most abundantly expressed neuregulins in the cerebral cortex during the period of synaptogenesis (*Fazzari et al., 2010*; *Longart et al., 2004*; *Rahman et al., 2019*). Previous studies have suggested that both Nrg1 and Nrg3 are homogeneously distributed in the same subcellular compartments. Nrg1 has been reported to be present in the axons of peripheral and hippocampal neurons in vitro (*Hancock et al., 2008*; *Vullhorst et al., 2017*; *Wolpowitz et al., 2000*), while Nrg3 has been shown to be enriched in axonal varicosities and synaptic puncta contacting interneuron dendrites (*Müller et al., 2018*; *Vullhorst et al., 2017*). Here, we demonstrate that sorting of Nrg1 and Nrg3 into different subcellular compartments of pyramidal cells mediate the formation of specific excitatory and inhibitory synapses in distinct populations of cortical interneurons. Our results unveil a novel strategy for the assembly of cortical circuits mediated by the subcellular sorting of family-related synaptic proteins.

## Results

### Distinct synaptic deficits in cortical pyramidal cells lacking Nrg1 or Nrg3

A large proportion of pyramidal cells express *Nrg1* and *Nrg3* across cortical layers 2–6 during postnatal development and into adulthood (*Figure 1—figure supplement 1*), suggesting that both neuregulins may play redundant functions as ErbB4 ligands in the regulation of inhibitory and excitatory synapses. To test this hypothesis, we performed genetic loss-of-function experiments using conditional alleles for each of these genes. Since neuregulin signaling in pyramidal cells plays a role in the laminar allocation of neocortical interneurons (*Bartolini et al., 2017*; *Flames et al., 2004*), we deleted *Nrg1* and *Nrg3* from pyramidal cells postnatally to avoid interfering with the function of neuregulins prior to synapse formation. To this end, we bred *Neurod6^{CreERT2}* (also known as *Nex^{CreERT2}*) mice, in which a tamoxifen-inducible version of Cre recombinase is under the control of the endogenous regulatory sequences of the pyramidal cell-specific *Neurod6* locus (*Agarwal et al., 2012*), with mice carrying loxP-flanked *Nrg1* or *Nrg3* alleles (*Bartolini et al., 2017*; *Yang et al., 2001*). Recombination of the *Nrg1* locus generates a *Nrg1* null allele in which the EGF domain is missing from all *Nrg1* variants (*Yang et al., 2001*). Since the precise time of full protein depletion from the neuron and therefore any impact in layer migration is difficult to predict, we first assessed Parvalbumin-expressing (PV+) interneuron distribution. We observed that postnatal induction of Cre recombinase in pyramidal cells does not disrupt the density and organization of PV+ interneurons in *Nrg1* and *Nrg3* conditional mutant mice at postnatal (P) day 30 (*Figure 1—figure supplement 2*). Therefore, these mice represent ideal models to investigate the precise role of *Nrg1* or *Nrg3* in pyramidal cells during the assembly of cortical circuits.

We first analyzed the role of *Nrg1* and *Nrg3* in the development of inhibitory synapses onto pyramidal cells. We investigated the two types of inhibitory synapses that are known to be altered in the

absence of ErbB4 from interneurons: synapses made by Cholecystokinin (CCK+) basket cells onto the soma and synapses made by chandelier cells on the axon initial segment (AIS) (*Del Pino et al., 2017*; *Del Pino et al., 2013*). We performed immunohistochemistry for pre- and postsynaptic markers to quantify inhibitory synaptic clusters in control and conditional mutant mice. In brief, we used GAD65 and CB1R to label the boutons of CCK+ interneurons, and GAD67 to identify chandelier cell boutons on the AIS of pyramidal cells (*Fish et al., 2011*; *Katona et al., 1999*). Of note, our ability to detect synaptic clusters was very similar independently of the genotype, which indicated that our approach does not introduce any bias in the quantification of synaptic densities (*Figure 1—figure supplement 3*). To identify Cre-expressing pyramidal cells, we included the conditional reporter allele *Ai9* (tdTomato) in our breeding scheme (*Madisen et al., 2010*). We observed a significant decrease in the density of GAD65+/CB1R+ presynaptic boutons contacting the soma of pyramidal cells in conditional *Nrg1* mutants compared to controls (*Figure 1A–C*). In contrast, deletion of *Nrg3* from pyramidal cells did not affect the formation of somatic synapses by CCK+ basket cells (*Figure 1A–C*). These results suggested that Nrg1, and not Nrg3, is required for the development of CCK+ basket cell synapses on pyramidal cells.

We next examined the function of neuregulins in the formation of inhibitory synapses by chandelier cells. We observed that the AIS of cortical pyramidal cells lacking *Nrg1* received significantly fewer GAD67+ boutons (*Figure 1—figure supplement 4*). In contrast, we observed that the density of axo-axonic boutons is unaltered following deletion of *Nrg3* from pyramidal cells (*Figure 1—figure supplement 4*). These results reinforced the idea that Nrg3 function is dispensable for the development of inhibitory synaptic inputs on pyramidal cells, a function that is mediated by Nrg1.

To establish whether Nrg1 functions in specific inhibitory circuits that are dependent on ErbB4 function (chandelier and CCK+ basket cell synapses) or is generally required for GABAergic synaptogenesis, we analyzed the number of PV+ basket cell synapses contacting pyramidal cells lacking specific neuregulins. We used Synaptotagmin-2 (Syt2) to specifically identify the presynaptic compartment of these synapses (*Sommeijer and Levelt, 2012*). We found no differences in the density of Syt2+ boutons or Syt2+/Gephyrin+ synaptic puncta contacting the soma of pyramidal cells lacking *Nrg1* compared to controls (*Figure 1—figure supplement 5*). These results are consistent with the hypothesis that Nrg1 plays a predominant role in the formation of inhibitory connections between specific types of interneurons (chandelier cells and CCK+ basket cells) and pyramidal cells. In addition, as expected from our previous results which suggest that Nrg3 is not involved in inhibitory synapse formation, we also observed that pyramidal cells lacking *Nrg3* receive a normal complement of PV+ basket cell synapses (*Figure 1—figure supplement 5*).

We next investigated the potential role of neuregulins in the assembly of excitatory synapses onto PV+ interneurons, since ErbB4 is located at these synapses and is essential for their formation (*Del Pino et al., 2017*; *Del Pino et al., 2013*; *Fazzari et al., 2010*; *Ting et al., 2011*). To this end, we selected regions of interest (ROIs) within layer 2/3 of the somatosensory cortex of control and neuregulin conditional mutant mice that exhibit comparable densities of tdTomato+ pyramidal cells (*Figure 1—figure supplement 6*). To identify excitatory synapses, we used vesicular glutamate transporter 1 (VGlut1), a characteristic component of excitatory glutamatergic terminals, and PSD95, the major scaffolding protein in the excitatory postsynaptic density. We quantified the density of VGlut1+/PSD95+ synapses within the population of tdTomato-expressing axonal terminals contacting PV+ interneurons. We found that conditional deletion of *Nrg1* from pyramidal cells does not alter the density of excitatory synapses these neurons make onto PV+ interneurons (*Figure 1A,D–E*). In contrast, we observed a significant reduction in the density of excitatory synapses targeting PV+ interneurons when pyramidal cells lacked *Nrg3* (*Figure 1A,D–E*). Of note, quantification of VGlut1+/PSD95+ synapses that did not contain tdTomato (i.e. arising from non-recombined, wild-type pyramidal neurons) revealed comparable values between control and conditional mutant mice (*Figure 1—figure supplement 7*), which reinforced the notion that synaptic deficits in conditional mutants are cell-autonomous.

Altogether, these genetic experiments reveal that Nrg1 and Nrg3 control inhibitory (inputs) and excitatory (outputs) synapse formation independently in pyramidal cells. Nrg1 mediates inhibitory synapse formation onto pyramidal cells, whereas Nrg3 controls excitatory synapse formation onto interneurons.

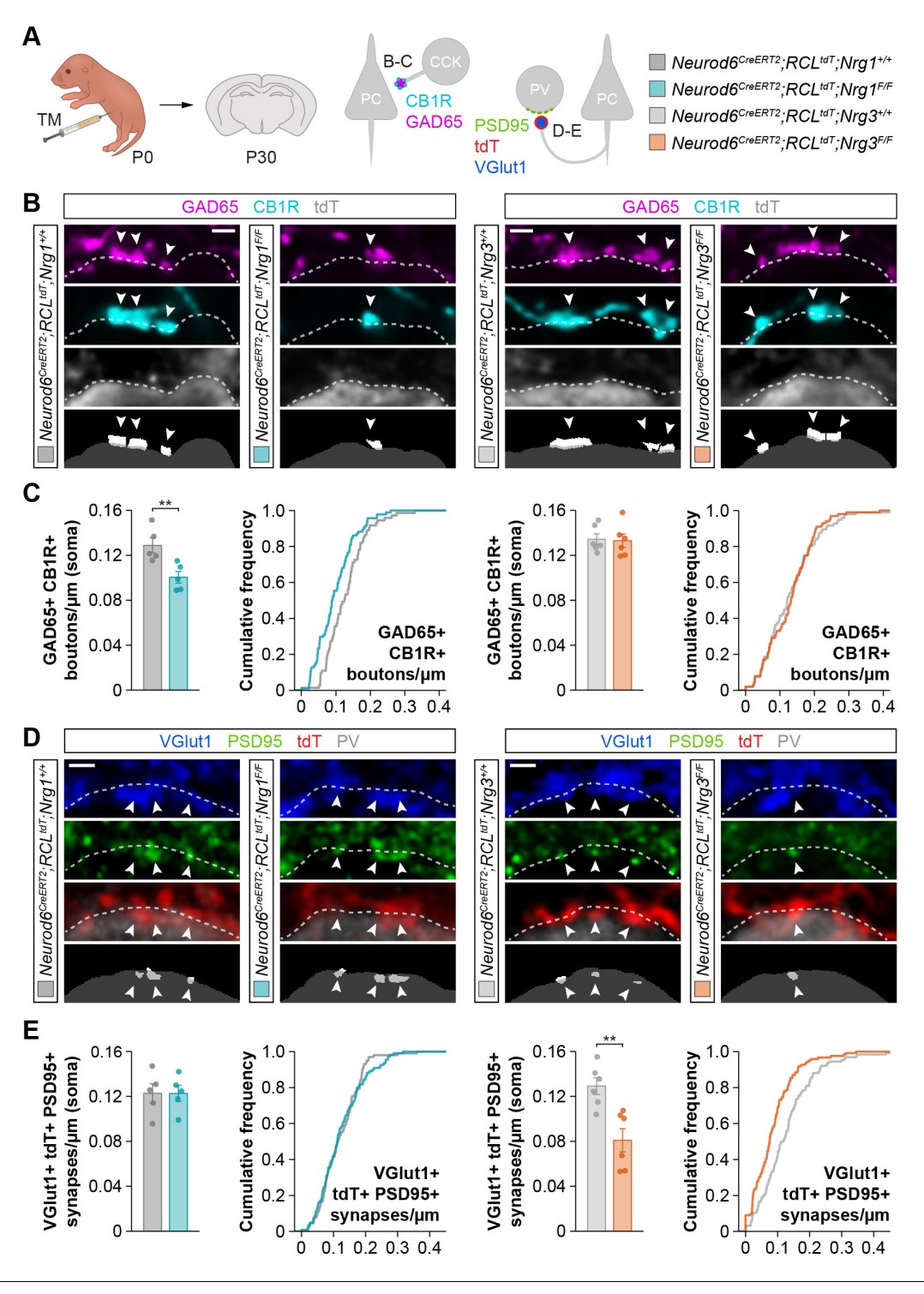

**Figure 1.** Specific synaptic deficits in cortical pyramidal cells lacking Nrg1 or Nrg3. (**A**) Schematic of experimental design. Conditional deletion of *Nrg1* or *Nrg3* in pyramidal cells was achieved by tamoxifen (TM) injection in newly born pups at P0. Two synapses were analyzed in conditional neuregulin mutants: inhibitory synapses formed by CCK+ basket cells onto the soma of pyramidal cells (GAD65, CB1R) and excitatory synapses made by pyramidal cells onto PV+ interneurons (VGlut1, PSD95). (**B**) Confocal images (top three panels) and binary images (bottom panel) illustrating presynaptic GABAergic boutons co-labeled with GAD65 (magenta) and CB1R (cyan) contacting the soma of tdTomato+ pyramidal cells (gray) in controls, *Nrg1* and *Nrg3* conditional mutant mice. (**C**) Quantification of the density of GAD65+/CB1R+ boutons formed onto pyramidal cell somas in *Nrg1* and *Nrg3*

*Figure 1 continued on next page*

*Figure 1 continued*

conditional mutant mice. Two-tailed Student's *t*-tests. For *Neurod6$^{CreERT2}$;RCL$^{tdt}$;Nrg1*, \*\*p<0.01; *n* = 5 mice (73 cells) for controls and 5 mice (94 cells) for mutants. For *Neurod6$^{CreERT2}$;RCL$^{tdt}$;Nrg3*, p=0.867; *n* = 6 mice (96 cells) for controls and 6 mice (100 cells) for mutants. (D) Confocal images (top three panels) and binary images (bottom panel) illustrating presynaptic VGlut1+ puncta (blue) in tdTomato+ axons (red) of pyramidal cells located in close apposition to PSD95+ clusters (green) in PV+ interneurons (gray) in controls, *Nrg1* and *Nrg3* conditional mutant mice. (E) Quantification of the density of VGlut1+/PSD95+/tdTomato+ synapses contacting PV+ interneurons in *Nrg1* and *Nrg3* conditional mutant mice. Two-tailed Student's *t*-tests. For *Neurod6$^{CreERT2}$;RCL$^{tdt}$;Nrg1*, p=0.993; *n* = 5 mice (101 cells) for controls and 5 mice (110 cells) for mutants. For *Neurod6$^{CreERT2}$;RCL$^{tdt}$;Nrg3*, \*\*p<0.01; *n* = 6 mice (126 cells) for controls and 6 mice (123 cells) for mutants. Scale bars, 1 μm. Data represent mean ± s. e.m. The averages per animal and genotype are represented in bar graphs, and the distributions of values per cell are shown in cumulative frequency plots. Data used for quantitative analyses are available in *Figure 1—source data 1* and *Figure 1—source data 2*.

The online version of this article includes the following source data and figure supplement(s) for figure 1:

**Source data 1.** Numerical data of inhibitory and excitatory synapses in conditional mutant mice for Nrg1 and Nrg3.

**Source data 2.** Summary of data and statistics that are represented in graphs.

**Figure supplement 1.** Cellular expression of *Nrg1* and *Nrg3* mRNA in the mouse cerebral cortex, related to *Figure 1*.

**Figure supplement 2.** Postnatal deletion of *Nrg1* or *Nrg3* from cortical pyramidal cells does not affect the density and distribution of PV+ interneurons, related to *Figure 1*.

**Figure supplement 2—source data 1.** Numerical data of PV+ cell densities in conditional mutant mice for Nrg1 and Nrg3.

**Figure supplement 3.** Extended view of synaptic labeling and analysis in brain slices from wild-type and conditional mutant mice.

**Figure supplement 3—source data 1.** Numerical data of synaptic colocalizations in conditional mutant mice for Nrg1 and Nrg3.

**Figure supplement 4.** Loss of axo-axonic boutons in cortical pyramidal cells lacking *Nrg1* but not *Nrg3*, related to *Figure 1*.

**Figure supplement 4—source data 1.** Numerical data of axo-axonic synapses in conditional mutant mice for Nrg1 and Nrg3.

**Figure supplement 5.** Density of inhibitory somatic synapses formed by PV+ basket cells onto pyramidal cells in conditional mutant mice for *Nrg1* or *Nrg3*, related to *Figure 1*.

**Figure supplement 5—source data 1.** Numerical data of PV+ basket cell synapses in conditional mutant mice for Nrg1 and Nrg3.

**Figure supplement 6.** Density of tamoxifen-induced recombined pyramidal cells, related to *Figure 1*.

**Figure supplement 6—source data 1.** Numerical data of tdT+ cell densities in conditional mutant mice for Nrg1 and Nrg3.

**Figure supplement 7.** Cell-autonomous requirement of Nrg3 in excitatory synapse formation, related to *Figure 1*.

**Figure supplement 7—source data 1.** Numerical data of tdT-negative excitatory synapses in conditional mutant mice for Nrg1 and Nrg3.

## Gain-of-function experiments support independent roles for Nrg1 and Nrg3

To strengthen the interesting observation of specific and non-overlapping synaptic deficits in *Nrg1* and *Nrg3* conditional mutant mice, we performed gain-of-function experiments by electroporating plasmids encoding *Nrg1* or *Nrg3* into pyramidal cell progenitors in embryonic day (E) 14.5 mouse embryos. We used Type III Neuregulin 1 (hereafter referred to as Nrg1), because it is the most abundant Nrg1 isoform in the cerebral cortex during synaptogenesis (*Fazzari et al., 2010*; *Longart et al., 2004*). Pyramidal neurons born from the electroporated progenitor cells were labeled by GFP, which was expressed from the same plasmids (*Figure 2A*). We observed that overexpression of Nrg1 or Nrg3 in embryonic pyramidal cells did not impair their migration and allocation into the cortex at P30 (data not shown). We next quantified the densities of inhibitory and excitatory synapses made onto or by GFP+ pyramidal cells at P30. To this end, we selected ROIs within layer 2/3 of the somatosensory cortex of electroporated mice that exhibit comparable densities of GFP+ pyramidal cells (*Figure 2—figure supplement 1*). Of note, our ability to detect synaptic clusters via immunofluorescence was comparable among the different experimental conditions (*Figure 2—figure supplement 2*). We observed that overexpression of Nrg1 resulted in a significant

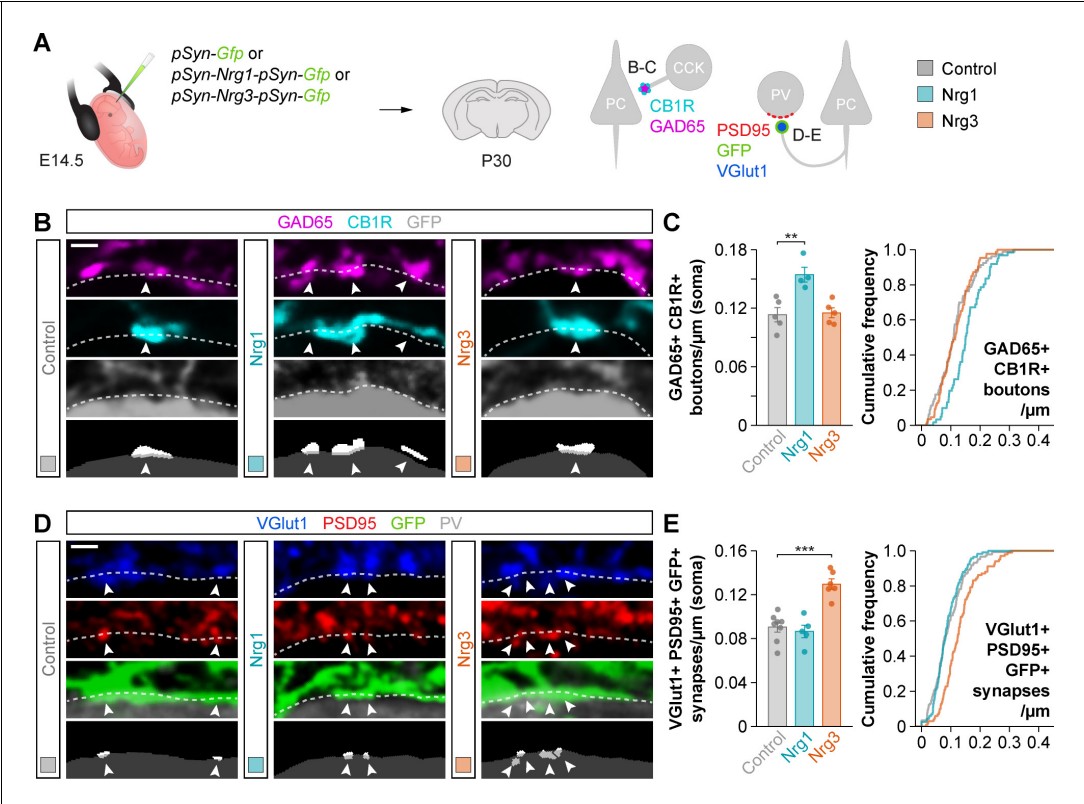

**Figure 2.** Gain-of-function experiments show the specificity of Nrg1 and Nrg3 in inhibitory and excitatory synapse formation. (A) Schematic of experimental design. In utero electroporation of *pSyn-Gfp* (control), *pSyn-Nrg1-pSyn-Gfp*, or *pSyn-Nrg3-pSyn-Gfp* plasmids into pyramidal cell progenitors was performed at E14.5 mouse embryos, and the density of synapses was analyzed in cortical superficial layers at P30: inhibitory synapses formed by CCK+ basket cells onto the soma of pyramidal cells (GAD65, CB1R), and excitatory synapses made by pyramidal cells onto PV+ interneurons (VGlut1, PSD95). (B) Confocal images (top three panels) and binary images (bottom panel) illustrating presynaptic GABAergic boutons co-labeled with GAD65 (magenta) and CB1R (cyan) contacting the soma of GFP+ pyramidal cells (gray) in electroporated mice. (C) Quantification of the density of GAD65+/CB1R+ boutons formed onto GFP+ pyramidal cell somas in gain-of-function experiments. One-way ANOVA: *F* = 11.100, p<0.01. Tukey's range test for post hoc comparison between control and experimental groups: for *pSyn-Nrg1-pSyn-Gfp*, **p<0.01; for *pSyn-Nrg3-pSyn-Gfp*, p=0.976; n = 5 mice (80 cells) for *pSyn-Gfp* (control), 4 mice (61 cells) for *pSyn-Nrg1-pSyn-Gfp*, and 5 mice (87 cells) for *pSyn-Nrg3-pSyn-Gfp*. (D) Confocal images (top three panels) and binary images (bottom panel) illustrating presynaptic VGlut1+ puncta (blue) in GFP+ axons (green) of pyramidal cells located in close apposition to PSD95+ clusters (red) in PV+ interneurons (gray) in electroporated mice. (E) Quantification of the density of VGlut1+/PSD95+/GFP+ synapses contacting PV+ interneurons in gain-of-function experiments. One-way ANOVA: *F* = 22.120, p<0.001. Tukey's range test for post hoc comparison between control and experimental groups: for *pSyn-Nrg1-pSyn-Gfp*, p=0.999; for *pSyn-Nrg3-pSyn-Gfp*, ***p<0.001; n = 8 mice (147 cells) for *pSyn-Gfp*, 5 mice (116 cells) for *pSyn-Nrg1-pSyn-Gfp*, and 6 mice (101 cells) for *pSyn-Nrg3-pSyn-Gfp*. Scale bars, 1 µm (B, D). Data represent mean ± s.e.m. The averages per animal and genotype are represented in bar graphs, and the distributions of values per cell are shown in cumulative frequency plots. Data used for quantitative analyses are available in *Figure 2—source data 1*.

The online version of this article includes the following source data and figure supplement(s) for figure 2:

**Source data 1.** Numerical data of inhibitory and excitatory synapses in gain-of-function experiments.

**Figure supplement 1.** Density of electroporated cortical pyramidal cells in gain-of-function experiments, related to *Figure 2*.

**Figure supplement 1—source data 1.** Numerical data of GFP+ cell densities in gain-of-function experiments.

**Figure supplement 2.** Extended view of synaptic labelling and analysis in brain slices from wild-type and electroporated mice.

**Figure supplement 2—source data 1.** Numerical data of synaptic colocalizations in gain-of-function experiments.

increase of somatic GAD65+/CB1R+ boutons innervating pyramidal cells, while Nrg3 overexpression did not change the density of these synaptic inputs (*Figure 2B–C*). In contrast, axonal excitatory synapses that pyramidal cells form onto PV+ interneurons were specifically increased by overexpression of Nrg3, but not Nrg1 (*Figure 2D–E*). These experiments add further support to the notion that Nrg1 and Nrg3 function in cortical pyramidal cells to induce the formation of inhibitory and excitatory synapses, respectively.

## Differential subcellular localization of neuregulins in pyramidal cells

We hypothesized that Nrg1 and Nrg3 differentially control the development of inhibitory and excitatory synapses in cortical circuits because they are targeted to different subcellular compartments in pyramidal cells. To explore this possibility, we performed another series of in utero electroporation experiments in which we expressed HA-tagged constructs of Nrg1 and Nrg3 in pyramidal cells (*Figure 3A*) and monitored the localization of both proteins at specific synapses. We found that Nrg1 and Nrg3 exhibit differential subcellular localization in pyramidal cells: whereas Nrg1 is spatially restricted to the perisomatic compartment of pyramidal cells, Nrg3 is highly enriched in the neuropil (*Figure 3B–C*). Analysis of endogenous protein expression by immunohistochemistry, using *Nrg1* and *Nrg3* conditional mutant as controls, confirmed these findings. For example, we observed the highest density of Nrg1+ puncta in the stratum pyramidale—the targeting layer of inhibitory somatic synapses in the hippocampus—in close apposition to GAD65+ inhibitory boutons (*Figure 3—figure supplement 1*). In contrast, Nrg3+ clusters were found to be particularly abundant within VGlut1+ presynaptic boutons coating the somatodendritic compartment of PV+ interneurons both in the hippocampus and neocortex (*Figure 3—figure supplement 2*), a finding consistent with previous studies (*Müller et al., 2018*). Altogether, these results revealed that Nrg1 and Nrg3 are differentially distributed across distinct subcellular compartments in pyramidal cells.

We next explored the localization of Nrg1 and Nrg3 with markers of inhibitory and excitatory synapses. We observed that Nrg1 puncta colocalize in synaptic clusters with Gephyrin, a scaffolding protein present in the postsynaptic membrane of inhibitory synapses (*Figure 3D*). These Nrg1 puncta were often opposed to presynaptic markers of inhibitory cells, such as GAD65 and CB1R, targeting the soma of pyramidal cells, and Nrg1 puncta were also observed within the AIS (*Figure 3D* and *Figure 3—figure supplement 3*). In contrast, we found that Nrg3 colocalizes to clusters contacting PV+ interneurons and the excitatory presynaptic marker VGlut1 (*Figure 3D* and *Figure 3—figure supplement 4*). We also detected Nrg3 puncta opposed to clusters expressing PSD95 (*Figure 3D*). In addition, we observed that GFP+ axon terminals from electroporated layer 2/3 pyramidal cells express HA-tagged Nrg3 in their presynaptic boutons innervating layer 5 PV+ interneurons, a target of superficial pyramidal cells in the cerebral cortex (*Figure 3—figure supplement 5*). Altogether, our results revealed a striking segregation in the subcellular distribution of neuregulins in pyramidal cells: Nrg1 is enriched in the postsynaptic compartment of inhibitory synapses targeting the soma of pyramidal cells whereas Nrg3 is mostly restricted to excitatory presynaptic terminals contacting interneurons (*Figure 3E*).

## The EGF-like domain of neuregulins does not mediate synapse specificity

The signaling capacity of neuregulins depends on their extracellular EGF-like domain, which binds to their receptors to trigger several intracellular signaling cascades (*Mei and Xiong, 2008*). To test whether the EGF-like domain of Nrg1 and Nrg3 is involved in the synaptic specificity exhibited by these molecules, we generated HA-tagged constructs of Nrg1 and Nrg3 in which we swapped their EGF-like domains and performed in utero electroporation to express them in pyramidal cells. We observed no differences in the subcellular localization of these chimeric constructs compared to wild-type Nrg1 and Nrg3: Nrg1 carrying the EGF-like domain of Nrg3 (Nrg1$^{EGF:Nrg3}$) localized to the soma, whereas Nrg3 carrying the EGF-like domain of Nrg1 (Nrg3$^{EGF:Nrg1}$) localized to the neuropil (*Figure 4A–B*). Synaptic targeting of Nrg1$^{EGF:Nrg3}$ to inhibitory postsynapses and of Nrg3$^{EGF:Nrg1}$ to excitatory presynaptic boutons was similar to that observed with Nrg1 and Nrg3 wild-type constructs (*Figure 4—figure supplement 1* and *Figure 4—figure supplement 2*).

We then examined how expression of these constructs influenced inhibitory synaptic inputs and excitatory synaptic outputs in electroporated pyramidal cells. Consistent with its somatic localization, we found that the receptor-binding chimeric construct of Nrg1 (Nrg1$^{EGF:Nrg3}$) increased the density of CCK+ presynaptic boutons innervating the soma of GFP+ pyramidal cells, as visualized by GAD65/CB1R colocalization, in similar quantities than wild-type Nrg1 (*Figure 2C* and *Figure 4C–D*). However, overexpression of the receptor-binding chimeric construct of Nrg3 (Nrg3$^{EGF:Nrg1}$), which is efficiently targeted to the axonal compartment, did not result in significant changes in CCK+ bouton density (*Figure 4C–D*).

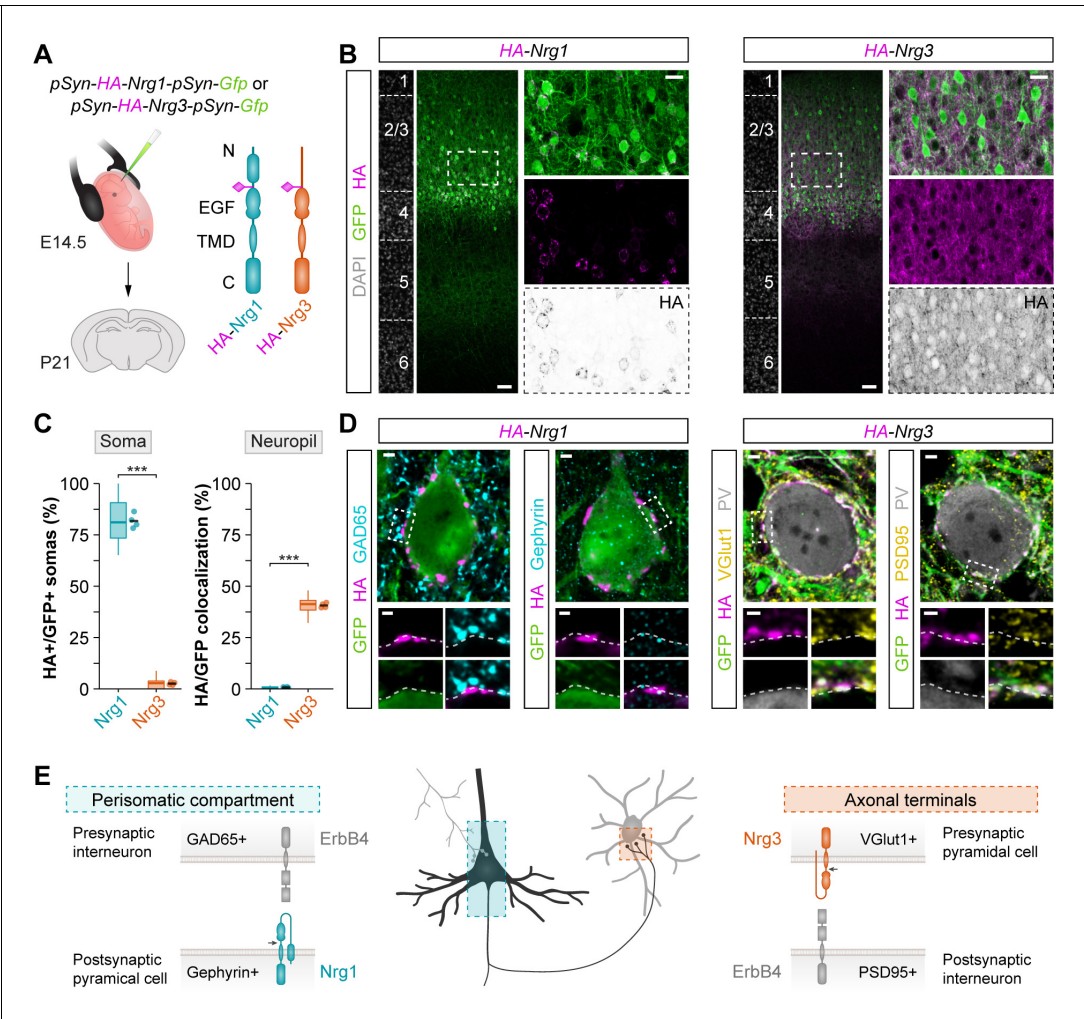

**Figure 3.** Nrg1 and Nrg3 have distinct subcellular distribution in pyramidal cells. (A) Schematic of experimental design. Plasmids encoding HA-tagged neuregulins and GFP were electroporated in pyramidal cell progenitors at E14.5, and the subcellular localization of neuregulins was analyzed at P21. The schematics depict the protein structure of neuregulins indicating the insertion of the HA tag in the extracellular. N, N-terminal domain; EGF, EGF-like domain; TMD, transmembrane domain; C, C-terminal domain. (B) Coronal sections through somatosensory cortex of P21 mice following in utero electroporation of *pSyn-Nrg1-pSyn-Gfp* or *pSyn-Nrg3-pSyn-Gfp* plasmids at E14.5. Sections were processed for immunohistochemistry against GFP (green) and HA (magenta) and counterstained with DAPI (gray). The high magnification images illustrate the localization of HA-tagged neuregulins in GFP+ pyramidal cells. The bottom panel depicts HA staining in a color-inverted image. Dotted squares indicate the localization of the cells shown in the high magnification images. (C) Quantification of the localization of HA+ neuregulin in the soma and neuropil of GFP+ pyramidal cells. Soma: Two-tailed Student's *t*-test, ***p<0.001; *n* = 4 mice (32 regions of interest, ROIs) for *pSyn-Nrg1-pSyn-Gfp*, *n* = 4 mice (32 ROIs) for *pSyn-Nrg3-pSyn-Gfp*. Neuropil: Two-tailed Student's *t*-test, ***p<0.001; *n* = 4 mice (32 ROIs) for *pSyn-Nrg1-pSyn-Gfp*, *n* = 4 mice (32 ROIs) for *pSyn-Nrg3-pSyn-Gfp*. (D) Localization of neuregulins with synaptic markers. Somatic Nrg1+ clusters (magenta) co-localize with Gephyrin (cyan) in the soma of pyramidal cells (green) in close proximity to presynaptic GABAergic boutons co-labeled by GAD65 (cyan). Nrg3+ clusters (magenta) co-localize with VGlut1+ (yellow) in presynaptic GFP+ axon terminals of pyramidal cells in close proximity to postsynaptic PSD95 clusters (yellow) in PV+ interneurons (gray). Dotted squares indicate the localization of the synaptic puncta shown in the high-magnification images. (E) Schematic illustrating the subcellular localization of Nrg1, Nrg3, and ErbB4 in cortical circuits. Scale bars, 50 μm (B) and 20 μm (high magnifications), and 2 μm (D) and 1 μm (high magnifications). Data from the distributions of ROIs are shown as box plots, and the adjacent data points and lines represent the averages per animal and averaged mean per group, respectively. Data used for quantitative analyses are available in *Figure 3—source data 1*.

The online version of this article includes the following source data and figure supplement(s) for figure 3:

**Source data 1.** Numerical data of subcellular localization of HA-tagged neuregulin constructs in electroporated pyramidal cells.
**Figure supplement 1.** Localization of endogenous Nrg1 in somas of cortical neurons, and specific targeting to inhibitory GABAergic clusters.
**Figure supplement 1—source data 1.** Numerical data of puncta density of endogenous Nrg1 protein in the cerebral cortex.
**Figure supplement 2.** Localization of endogenous Nrg3 in the neuropil of the neocortex, and specific targeting to excitatory presynaptic boutons innervating PV+ interneurons.
**Figure supplement 2—source data 1.** Numerical data of puncta density of endogenous Nrg3 protein in the cerebral cortex.
*Figure 3 continued on next page*

*Figure 3 continued*

**Figure supplement 3.** Subcellular localization and targeting of HA-tagged Nrg1 to inhibitory perisomatic inputs of cortical pyramidal cells.

**Figure supplement 3—source data 1.** Numerical data of synaptic targeting of HA-tagged neuregulin constructs in electroporated (GFP+) and non-electroporated (GFP–) pyramidal cells.

**Figure supplement 4.** Synaptic targeting of wild-type neuregulin constructs to excitatory presynaptic boutons innervating PV+ interneurons.

**Figure supplement 4—source data 1.** Numerical data of synaptic targeting of HA-tagged neuregulin constructs in axon terminals innervating PV+ interneurons.

**Figure supplement 5.** Targeting of HA-tagged Nrg3 to axon terminals innervating PV+ interneurons in layer 5 of the cerebral cortex.

**Figure supplement 5—source data 1.** Numerical data of neuropil colocalization of HA-tagged neuregulin constructs in layer 5.

**Figure supplement 5—source data 2.** Numerical data of synaptic targeting of HA-tagged neuregulin constructs in axon terminals innervating PV+ interneurons in layer 5.

**Figure supplement 6.** Synaptic targeting of HA-tagged Nrg3 construct to presynaptic terminals that contact ErbB4 postsynaptic clusters in PV+ interneurons.

**Figure supplement 6—source data 1.** Numerical data of co-localization of HA-tagged Nrg3 construct and ErbB4+ clusters in PV+ interneurons.

To examine the synaptogenic properties of neuregulin chimeric constructs in the axonal compartment, we measured VGlut1+/PSD95+ synapse density within GFP+ pyramidal cell axons targeting PV+ interneurons. The receptor-binding chimeric construct of Nrg3 (Nrg3$^{EGF:Nrg1}$) significantly augmented the density of axonal excitatory synapses that electroporated pyramidal cells made onto PV+ interneurons, at comparable levels than wild-type Nrg3 (*Figure 2E* and *Figure 4E–F*). In contrast, overexpression of the receptor-binding chimeric construct of Nrg1 (Nrg1$^{EGF:Nrg3}$), which is targeted to the perisomatic region, did not change excitatory synapse densities in axon terminals (*Figure 4E–F*). Thus, chimeric neuregulins containing the EGF-like domain of the homologous neuregulin member can recapitulate the specific synaptic functions of wild-type Nrg1 and Nrg3 in vivo. These findings demonstrate that Nrg1 and Nrg3 function in inhibitory and excitatory synapse formation does not depend on their differential binding properties to their receptors, but on their selective sorting to the somatic and axonal compartment, respectively.

## Structural differences mediate the subcellular distribution of neuregulins

Since the extracellular EGF-like domain of Nrg1 and Nrg3 is not responsible for the differential functions of these two family-related synaptic molecules, we hypothesized that the intracellular region (C-terminal) of these proteins might be important for their localization and therefore differential physiological roles (*Figure 5—figure supplement 1*). To test this idea, we engineered HA-tagged constructs in which the intracellular domains (C-terminal) were swapped between both neuregulins and used in utero electroporation to express them in pyramidal cells. We observed that the C-terminal domain of Nrg3 was sufficient to re-route some Nrg1 from the soma to the neuropil (Nrg1$^{Ct:Nrg3}$; *Figure 5A–B*), although with limited efficiency compared to full-length Nrg3 (*Figure 3C*). Accordingly, the proportion of VGlut1+/GFP+ boutons innervating PV+ interneurons that express the chimeric Nrg1$^{Ct:Nrg3}$ protein was significantly reduced compared to Nrg3 wild-type protein, which suggested that this chimeric protein is less efficiently expressed and/or targeted to axonal excitatory synapses than the wild-type form (*Figure 5—figure supplement 2*). We also found that the intracellular domain of Nrg1 is sufficient to sequester a large fraction of Nrg3 in the soma, dramatically decreasing its normal targeting to the neuropil (Nrg3$^{Ct:Nrg1}$; *Figure 5A–B* and *Figure 5—figure supplement 2*). Remarkably, the chimeric Nrg3$^{Ct:Nrg1}$ proteins found in the soma of electroporated GFP+ pyramidal cells were precisely located in postsynaptic clusters adjacent to inhibitory inputs (*Figure 5—figure supplement 3*). These results suggest that the intracellular, C-terminal domains of Nrg1 and Nrg3 contain unique sequences that influence the subcellular trafficking of both neuregulins.

The previous experiments suggested that C-terminal domain-dependent subcellular sorting may determine the specificity of neuregulin signaling in inhibitory and excitatory synapse formation. To test this hypothesis, we first quantified the density of CCK+ somatic inputs in pyramidal cells expressing the C-terminal domain-swapping neuregulin proteins. Consistent with its inefficient retention in the soma (*Figure 5A–B* and *Figure 5—figure supplement 3*), we observed that overexpression of a chimeric Nrg1 protein containing the C-terminal domain of Nrg3 (Nrg1$^{Ct:Nrg3}$) does not

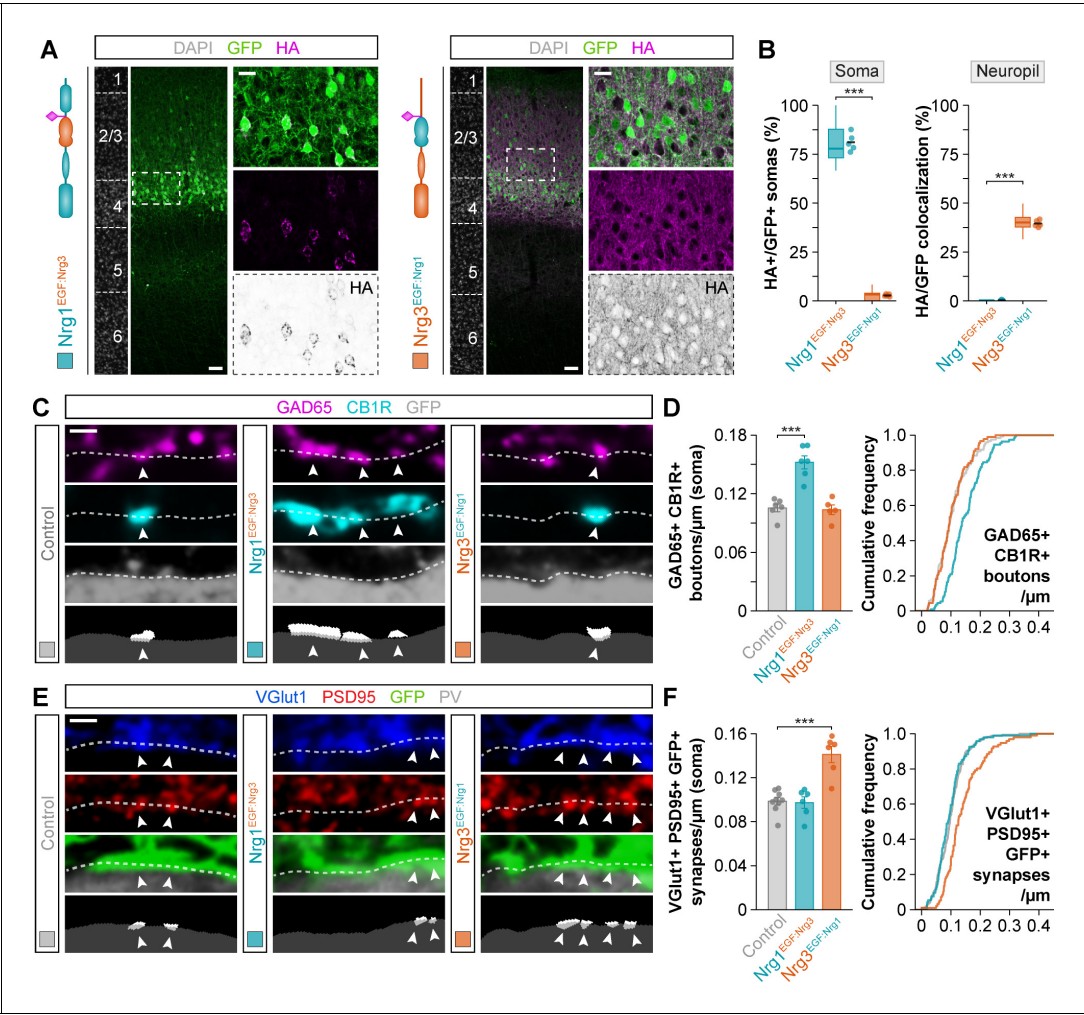

**Figure 4.** Neuregulin-dependent synapse formation and localization is not specified by the EGF-like domain. (**A**) Coronal sections through somatosensory cortex of P30 mice following in utero electroporation of *pSyn-Nrg1$^{EGF:Nrg3}$-pSyn-Gfp* or *pSyn-Nrg3$^{EGF:Nrg1}$-pSyn-Gfp* plasmids at E14.5. Sections were processed for immunohistochemistry against GFP (green) and HA (magenta) and counterstained with DAPI (gray). The high-magnification images illustrate the localization of HA-tagged neuregulins in GFP+ pyramidal cells. The bottom panel depicts HA staining in a color-inverted image. Dotted squares indicate the localization of the cells shown in the high-magnification images. The schematics illustrate the structure of chimeric neuregulins in which the EGF-like domain was swapped between Nrg1 and Nrg3. (**B**) Quantification of the localization of HA+ neuregulin in the soma and neuropil of GFP+ pyramidal cells. Soma: Two-tailed Student's *t*-test, ***p<0.001; *n* = 5 mice (40 regions of interest, ROIs) for *pSyn-Nrg1$^{EGF:Nrg3}$-pSyn-Gfp*, *n* = 5 mice (40 ROIs) for *pSyn-Nrg3$^{EGF:Nrg1}$-pSyn-Gfp*. Neuropil: Two-tailed Student's *t*-test, ***p<0.001; *n* = 5 mice (40 ROIs) for *pSyn-Nrg1$^{EGF:Nrg3}$-pSyn-Gfp*, *n* = 5 mice (40 ROIs) for *pSyn-Nrg3$^{EGF:Nrg1}$-pSyn-Gfp*. (**C**) Confocal images (top three panels) and binary images (bottom panel) illustrating presynaptic boutons co-labeled with GAD65 (magenta) and CB1R (cyan) innervating the soma of GFP+ pyramidal cells (gray) in EGF-like domain swapping experiments. (**D**) Quantification of the density of GAD65+/CB1R+ boutons contacting GFP+ pyramidal cells in gain-of-function chimera experiments. One-way ANOVA: *F* = 26.790, p<0.001. Tukey's range test for post hoc comparison between control and experimental groups: for *pSyn-Nrg1$^{EGF:Nrg3}$-pSyn-Gfp*, ***p<0.001; for *pSyn-Nrg3$^{EGF:Nrg1}$-pSyn-Gfp*, p=0.968; n = 6 mice (120 cells) for *pSyn-Gfp* (control), 6 mice (110 cells) for *pSyn-Nrg1$^{EGF:Nrg3}$-pSyn-Gfp*, 5 mice (88 cells) for *pSyn-Nrg3$^{EGF:Nrg1}$-pSyn-Gfp*. (**E**) Confocal images (top three panels) and binary images (bottom panel) illustrating presynaptic VGlut1+ puncta (blue) in GFP+ axons (green) of pyramidal cells located in close apposition to PSD95+ clusters (red) in PV+ interneurons (gray) in EGF-like domain swapping experiments. (**F**) Quantification of the density of VGlut1+/PSD95+/GFP+ synapses contacting PV+ interneurons in gain-of-function chimera experiments. One-way ANOVA: *F* = 21.820, p<0.001. Tukey's range test for post hoc comparison between control and experimental groups: for *pSyn-Nrg1$^{EGF:Nrg3}$-pSyn-Gfp*, p=0.982, for *pSyn-Nrg3$^{EGF:Nrg1}$-pSyn-Gfp*, ***p<0.001; n = 9 mice (167 cells) for *pSyn-Gfp* (control), 6 mice (94 cells) for *pSyn-Nrg1$^{EGF:Nrg3}$-pSyn-Gfp*, 6 mice (96 cells) for *pSyn-Nrg3$^{EGF:Nrg1}$-pSyn-Gfp*. Data from the distributions of ROIs are shown as box plots, and the adjacent data points and lines represent the averages per animal and averaged mean per group, respectively. Scale bars, 50 μm (**A**) and 20 μm (high magnification), and 1 μm (**C, E**). Data in synaptic quantifications represent mean ± s.e.m. The averages per animal and electroporation condition are represented in bar graphs, and the distributions of values per cell are shown in cumulative frequency plots. Data used for quantitative analyses are available in *Figure 4—source data 1*, and *Figure 4—source data 2*.

The online version of this article includes the following source data and figure supplement(s) for figure 4:

*Figure 4 continued on next page*

*Figure 4 continued*

**Source data 1.** Numerical data of subcellular localization in gain-of-function experiments with EGF-like domain chimeric neuregulin constructs.
**Source data 2.** Numerical data of inhibitory and excitatory synapses in gain-of-function experiments with EGF-like domain chimeric neuregulin constructs.
**Figure supplement 1.** Synaptic targeting of EGF-like domain-swapping neuregulin constructs to inhibitory postsynaptic clusters in the somatic compartment of cortical pyramidal cells.
**Figure supplement 1—source data 1.** Numerical data of synaptic targeting of EGF-like domain chimeric neuregulin constructs in GFP+ electroporated pyramidal cells.
**Figure supplement 2.** Targeting of EGF-like domain-swapping neuregulin constructs to excitatory presynaptic boutons innervating PV+ interneurons.
**Figure supplement 2—source data 1.** Numerical data of synaptic targeting of EGF-like domain chimeric neuregulin constructs in axon terminals innervating PV+ interneurons.

increase the number of CB1R+/GAD65+ presynaptic boutons contacting electroporated pyramidal cells (*Figure 5C–D*). In contrast, overexpression of a chimeric Nrg3 protein carrying the C-terminal domain of Nrg1 (Nrg3$^{Ct:Nrg1}$), which is abnormally retained in the soma of pyramidal cells (*Figure 5A–B* and *Figure 5—figure supplement 3*), led to a significant increase in the density of these GABAergic inputs (*Figure 5C–D*). Secondly, we measured the ability of these chimeric constructs to influence the formation of excitatory synapses onto PV+ interneurons. Consistent with their inefficient transport to the neuropil compared to wild-type Nrg3 (*Figure 5A–B* and *Figure 5—figure supplement 2*), neither Nrg1$^{Ct:Nrg3}$ nor Nrg3$^{Ct:Nrg1}$ caused a significant change in the number of VGlut1+/PSD95+ puncta within GFP+ axon terminals contacting PV+ interneurons (*Figure 5E–F*). These experiments confirmed that the efficient targeting of neuregulins to specific subcellular compartments in pyramidal cells mediates their function in synapse formation. They also revealed that the C-terminal domain of neuregulins is essential for the specificity of this process.

To add further support to this idea, we performed a final series of experiments in which we electroporated plasmids encoding truncated forms of Nrg1 and Nrg3 that lack the entire intracellular region, the C-terminal domain. We found that both Nrg1 and Nrg3 lacking the C-terminal domain (Nrg1$^{\Delta Ct}$ and Nrg3$^{\Delta Ct}$) lose the specific subcellular distribution found in wild-type Nrg1 and Nrg3 (*Figure 6A–B*). Specifically, we observed that Nrg1$^{\Delta Ct}$ was no longer restricted to the somatic compartment of pyramidal cells and was in turn abundantly found throughout the neuropil, both in dendrites and axon terminals (*Figure 6A–B*). On the other hand, we observed that Nrg3$^{\Delta Ct}$ was no longer restricted to axons as it is the case for wild-type Nrg3: it was also abundantly found in the somatodendritic compartment of pyramidal cells (*Figure 6A–B*). Consistently, both Nrg1$^{\Delta Ct}$ and Nrg3$^{\Delta Ct}$ were observed in somatic inhibitory synapses as well as axonal excitatory synapses in electroporated GFP+ pyramidal cells (*Figure 6—figure supplements 1–2*). This finding reinforced the idea that the specific localization of Nrg1 and Nrg3 in the somatic and axonal compartments, respectively, requires sequences encoded in their C-terminal domain.

We examined how the lack of specificity in the distribution of neuregulins would impact the formation of inhibitory inputs and excitatory outputs in pyramidal cells. First, we assessed the effect of Nrg1 and Nrg3 proteins lacking the C-terminal domain in inducing the formation of GABAergic boutons from CCK+ basket cells onto the soma of pyramidal neurons. Strikingly, overexpression of truncated forms of Nrg1 and Nrg3 resulted in a significant increase in the density of GAD65+/CB1R+ boutons innervating electroporated pyramidal cells (*Figure 6C–D*). Of note, these phenotypes appeared to be moderate as compared to overexpression experiments of wild-type Nrg1 protein (*Figure 2B–C*), probably due to the fact that the targeting of both neuregulins lacking the C-terminal domain to the somatic compartment is less efficient than the wild-type Nrg1 protein (*Figure 3B–C* and *Figure 6—figure supplement 1*). Second, we quantified excitatory synaptic inputs contacting PV+ interneurons from pyramidal cells expressing the truncated forms of Nrg1 and Nrg3. Overexpression of these truncated proteins robustly induced the formation of VGlut1+/PSD95+ synaptic contacts within GFP+ axon terminals innervating PV+ interneurons (*Figure 6E–F*). Altogether, the ability of neuregulins lacking their C-terminal domains to induce synapse formation in vivo suggests that their synaptogenic function does not depend on intracellular signaling pathways but rather on the role mediated by the extracellular domain at the synaptic membrane. These results are consistent with the hypothesis that the selective subcellular segregation of Nrg1 and Nrg3 in cortical pyramidal cells controls their specific function in inhibitory and excitatory synapses, respectively.

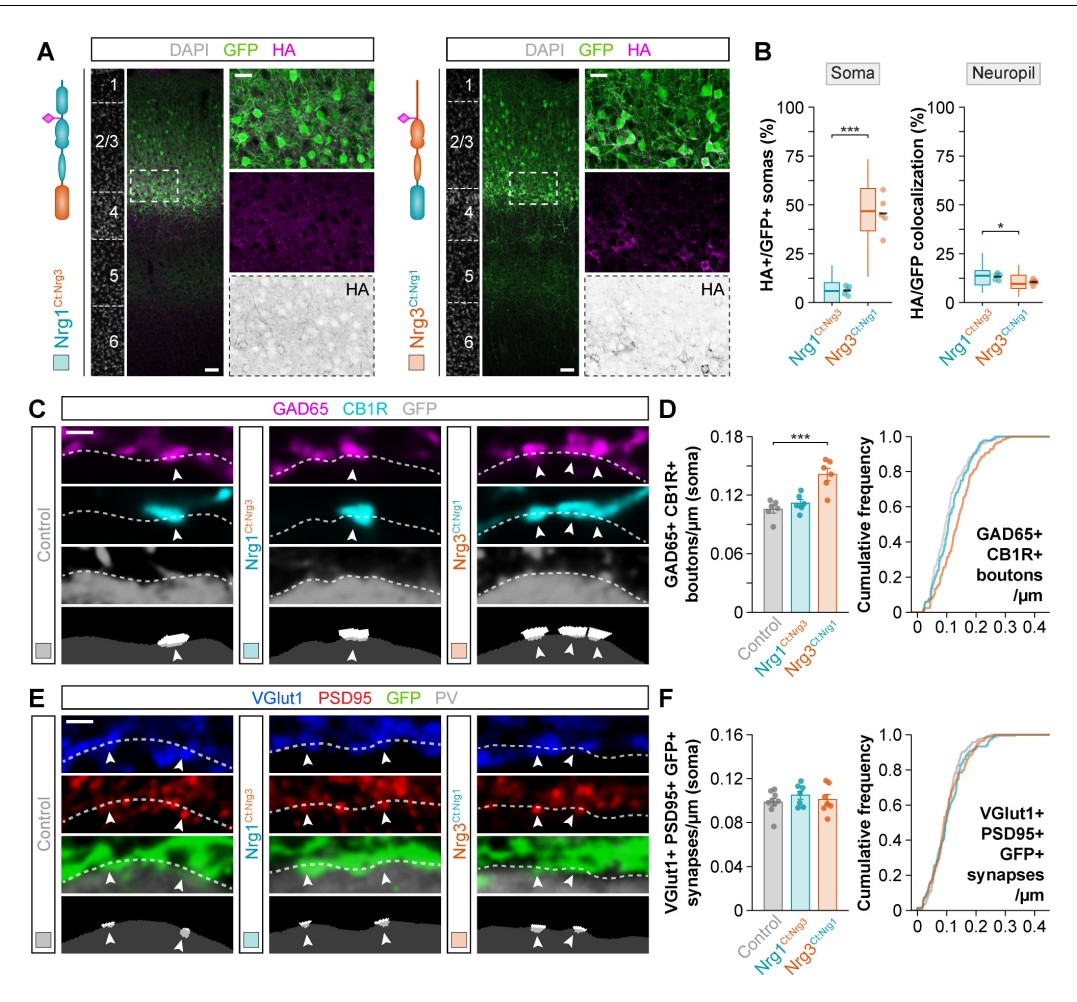

**Figure 5.** The C-terminal domain of Nrg1 and Nrg3 determines subcellular sorting to input and output synapses in pyramidal cells. (**A**) Coronal sections through somatosensory cortex of P30 mice following in utero electroporation of *pSyn-Nrg1^{Ct:Nrg3}-pSyn-Gfp* or *pSyn-Nrg3^{Ct:Nrg1}-pSyn-Gfp* plasmids at E14.5. Sections were processed for immunohistochemistry against GFP (green) and HA (magenta) and counterstained with DAPI (gray). The high-magnification images illustrate the localization of HA-tagged neuregulins in GFP+ pyramidal cells. The bottom panel depicts HA staining in a color-inverted image. Dotted squares indicate the localization of the cells shown in the high-magnification images. The schematics illustrate the structure of chimeric neuregulins in which the intracellular, C-terminal domain was swapped between Nrg1 and Nrg3. (**B**) Quantification of the localization of HA+ neuregulin in the soma and neuropil of GFP+ pyramidal cells. Soma: Two-tailed Student's *t*-test, ***p<0.001; *n* = 5 mice (40 regions of interest, ROIs) for *pSyn-Nrg1^{Ct:Nrg3}-pSyn-Gfp*, *n* = 5 mice (40 ROIs) for *pSyn-Nrg3^{Ct:Nrg1}-pSyn-Gfp*. Neuropil: Two-tailed Student's *t*-test, *p<0.05; *n* = 5 mice (40 ROIs) for *pSyn-Nrg1^{Ct:Nrg3}-pSyn-Gfp*, *n* = 5 mice (40 ROIs) for *pSyn-Nrg3^{Ct:Nrg1}-pSyn-Gfp*. (**C**) Confocal images (top three panels) and binary images (bottom panel) illustrating presynaptic boutons co-labeled with GAD65 (magenta) and CB1R (cyan) innervating the soma of GFP+ pyramidal cells (gray) in C-terminal domain swapping experiments. (**D**) Quantification of the density of GAD65+/CB1R+ boutons contacting GFP+ pyramidal cells in gain-of-function chimera experiments. One-way ANOVA: *F* = 15.640, p<0.001. Tukey's range test for post hoc comparison between control and experimental groups: for *pSyn-Nrg1^{Ct:Nrg3}-pSyn-Gfp*, p=0.632; for *pSyn-Nrg3^{Ct:Nrg1}-pSyn-Gfp*, ***p<0.001; n = 6 mice (120 cells) for *pSyn-Gfp* (control), 6 mice (136 cells) for *pSyn-Nrg1^{Ct:Nrg3}-pSyn-Gfp*, 6 mice (127 cells) for *pSyn-Nrg3^{Ct:Nrg1}-pSyn-Gfp*. (**E**) Confocal images (top three panels) and binary images (bottom panel) illustrating presynaptic VGlut1+ puncta (blue) in GFP+ axons (green) of pyramidal cells located in close apposition to PSD95+ clusters (red) in PV+ interneurons (gray) in C-terminal domain swapping experiments. (**F**) Quantification of the density of VGlut1+/PSD95+/GFP+ synapses contacting PV+ interneurons in gain-of-function chimera experiments. One-way ANOVA: *F* = 0.679, p=0.519. Tukey's range test for post hoc comparison between control and experimental groups: for *pSyn-Nrg1^{Ct:Nrg3}-pSyn-Gfp*, p=0.489, for *pSyn-Nrg3^{Ct:Nrg1}-pSyn-Gfp*, p=0.909; n = 9 mice (167 cells) for *pSyn-Gfp* (control), 7 mice (121 cells) for *pSyn-Nrg1^{Ct:Nrg3}-pSyn-Gfp*, 7 mice (135 cells) for *pSyn-Nrg3^{Ct:Nrg1}-pSyn-Gfp*. Scale bars, 50 μm (**A**) and 20 μm (high magnification), and 1 μm (**C, E**). Data from the distributions of ROIs are shown as box plots, and the adjacent data points and lines represent the averages per animal and averaged mean per group, respectively. Data in synaptic quantifications represent mean ± s.e.m. The averages per animal and electroporation condition are represented in bar graphs, and the distributions of values per cell are shown in cumulative frequency plots. Data used for quantitative analyses are available in *Figure 5—source data 1*, and *Figure 5—source data 2*.

The online version of this article includes the following source data and figure supplement(s) for figure 5:

*Figure 5 continued on next page*

*Figure 5 continued*

**Source data 1.** Numerical data of subcellular localization in gain-of-function experiments with C-terminal domain chimeric neuregulin constructs.

**Source data 2.** Numerical data of inhibitory and excitatory synapses in gain-of-function experiments with C-terminal domain chimeric neuregulin constructs.

**Figure supplement 1.** Comparison of amino acid sequences of the EGF-like domain and the C-terminal domains of Nrg1 and Nrg3.

**Figure supplement 2.** Targeting of C-terminal domain-swapping neuregulin constructs to excitatory presynaptic boutons innervating PV+ interneurons.

**Figure supplement 2—source data 1.** Numerical data of synaptic targeting of C-terminal domain chimeric neuregulin constructs in axon terminals innervating PV+ interneurons.

**Figure supplement 3.** Synaptic targeting of C-terminal domain-swapping neuregulin constructs to inhibitory postsynaptic clusters in the somatic compartment of cortical pyramidal cells.

**Figure supplement 3—source data 1.** Numerical data of synaptic targeting of C-terminal domain chimeric neuregulin constructs in GFP+ electroporated pyramidal cells.

## Discussion

Our findings unveil how subcellular sorting of synaptic proteins orchestrates the assembly of excitatory and inhibitory synapses in cortical circuits. Previous studies have shown that the tyrosine kinase receptor ErbB4 is required for the wiring of specific types of cortical interneurons, in which it contributes to the formation of both inhibitory outputs and excitatory inputs. Here we demonstrate that Nrg1 and Nrg3, two different members of the neuregulin family of trophic factors co-expressed by pyramidal cells, mediate the development of inhibitory synapses made and excitatory synapses received, respectively, by ErbB4-expressing cortical interneurons. Our study highlights the crucial role of polarized protein trafficking for synaptic specificity in the formation of neuronal circuits.

Proteins are synthesized in the rough endoplasmic reticulum, modified through the Golgi complex, and packaged into carrier vesicles for their transport to different subcellular domains. The efficient targeting of membrane proteins to distinct subcellular compartments relies on specific protein-protein interactions that regulate vesicle trafficking. In particular, membrane proteins are sorted into different pools of vesicles for selective delivery to the somatodendritic or axonal compartments while being transported through the trans-Golgi network (*Ribeiro et al., 2018*). Our study demonstrates that the C-terminal domains of Nrg1 and Nrg3 are essential for the sorting of these synaptic proteins into the somatic and axonal compartments, respectively (*Figure 5* and *Figure 6*). This is consistent with the existence of amino acid sequences in the cytoplasmic domain of other transmembrane proteins that are required for their accurate subcellular sorting (*Farías et al., 2012*; *Gu, 2003*; *Li et al., 2016*; *Rivera et al., 2003*). Our experiments also indicate that the C-terminal, intracellular domains of neuregulins are indispensable for the specific formation of inhibitory and excitatory synapses in pyramidal cells in vivo. Our data suggest that, while the C-terminal domain plays critical roles in specifying the selective subcellular localization of Nrg1 and Nrg3, their extracellular domain might be sufficient to bind and activate their receptor at the pre- and postsynaptic sites, respectively, to induce synaptogenesis. This is in contrast with the mechanisms of action of postsynaptic Neuroligin-1, whose varied functions at the synapse rely on signaling pathways associated with both its intra- and extracellular domains (*Wu et al., 2019*).

ErbB4 is the most likely receptor mediating the effect on Nrg1 and Nrg3 in the wiring of cortical interneurons. ErbB4 is a classical binding partner of both neuregulins (*Mei and Nave, 2014*) and is exclusively expressed in the postnatal cortex by several types of cortical interneurons (*Fazzari et al., 2010*; *Vullhorst et al., 2009*). Using electron microscopy, we have previously shown that ErbB4 is very abundant in the postsynaptic density of excitatory synapses received by interneurons (*Fazzari et al., 2010*). In addition, ErbB4 is also found, albeit at lower density (i.e. relatively fewer immunogold particles compared to those found in postsynaptic densities), in inhibitory axo-somatic and axo-axonic terminals. At the light microscopy level, Nrg3 and ErbB4 colocalize extensively in excitatory synaptic puncta contacting PV+ interneurons (*Figure 3—figure supplement 6*), which is consistent with a role for this ligand-receptor pair in the formation of these synapses. In contrast, the relatively low levels of ErbB4 in the axon terminals of CCK+ basket cells and chandelier cells prevent their detection at the light microscopy level (*Fazzari et al., 2010*), which precludes the assessment of Nrg1-ErbB4 colocalization in the same synapses. Although it is formally possible that Nrg1 function in inhibitory synapse formation is mediated by a yet unidentified receptor, the complementary phenotypes observed in conditional *Nrg1*, *Nrg3* and *ErbB4* mutants strongly suggests otherwise.

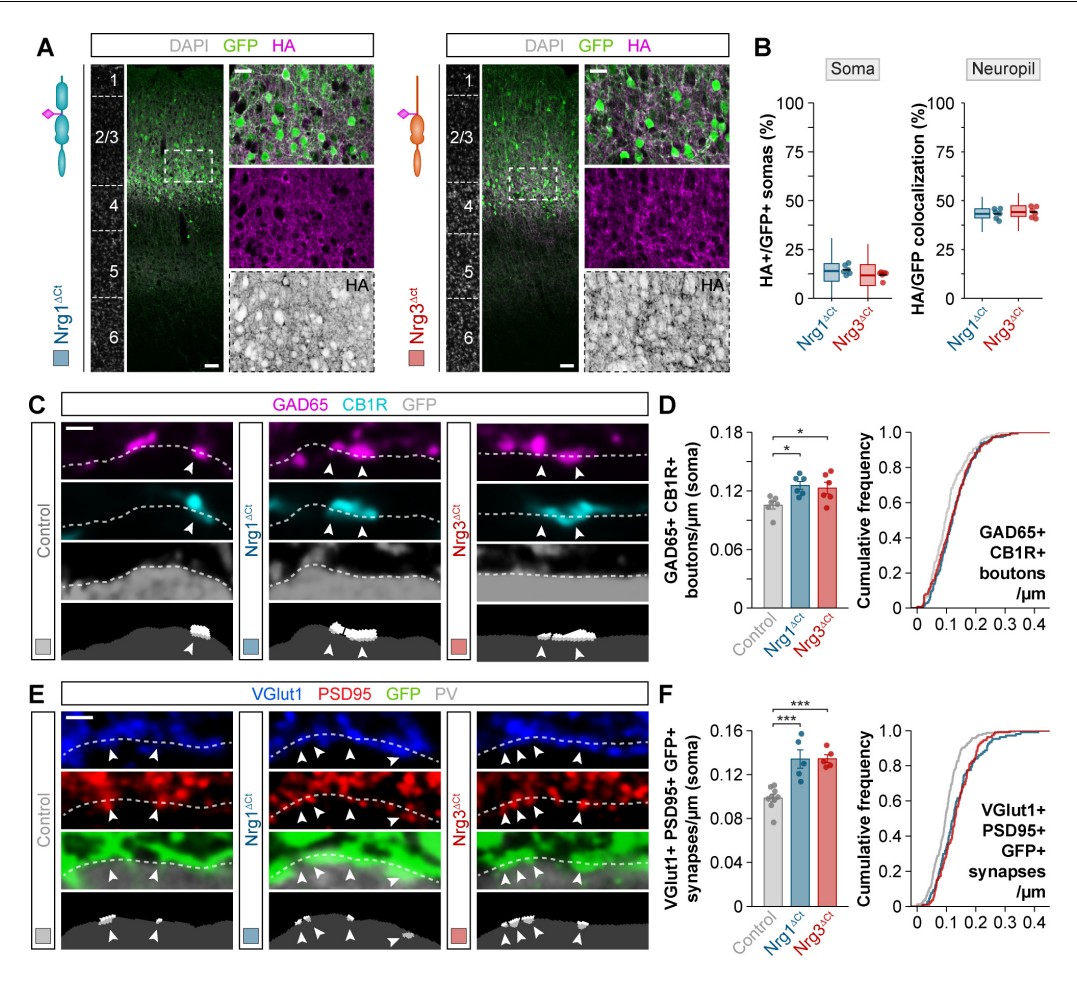

**Figure 6.** Subcellular segregation of neuregulins is encoded in the C-terminal domain. (**A**) Coronal sections through somatosensory cortex of P30 mice following in utero electroporation of *pSyn-Nrg1^{ΔCt}-pSyn-Gfp* or *pSyn-Nrg3^{ΔCt}-pSyn-Gfp* plasmids at E14.5. Sections were processed for immunohistochemistry against GFP (green) and HA (magenta) and counterstained with DAPI (gray). The high-magnification images illustrate the localization of HA-tagged neuregulins in GFP+ pyramidal cells. The bottom panel depicts HA staining in a color-inverted image. Dotted squares indicate the localization of the cells shown in the high-magnification images. The schematics illustrate the structure of truncated neuregulins in which the C-terminal domain was removed. (**B**) Quantification of the localization of HA+ neuregulin in the soma and neuropil of GFP+ pyramidal cells. Soma: Mann-Whitney *U*-test, p=0.093; n = 6 mice (66 regions of interest, ROIs) for *pSyn-Nrg1^{ΔCt}-pSyn-Gfp*, n = 6 mice (66 ROIs) for *pSyn-Nrg3^{ΔCt}-pSyn-Gfp*. Neuropil: Two-tailed Student's *t*-test, p=0.589; n = 6 mice (58 ROIs) for *pSyn-Nrg1^{ΔCt}-pSyn-Gfp*, n = 6 mice (58 ROIs) for *pSyn-Nrg1^{ΔCt}-pSyn-Gfp*. (**C**) Confocal images (top three panels) and binary images (bottom panel) illustrating presynaptic boutons co-labeled with GAD65 (magenta) and CB1R (cyan) innervating the soma of GFP+ pyramidal cells (gray) in C-terminal domain deletion experiments. (**D**) Quantification of the density of GAD65+/CB1R+ boutons contacting GFP+ pyramidal cells in gain-of-function truncation experiments. One-way ANOVA: *F* = 5.325, p<0.05. Tukey's range test for post hoc comparison between control and experimental groups: for *pSyn-Nrg1^{ΔCt}-pSyn-Gfp*, *p<0.05; for *pSyn-Nrg3^{ΔCt}-pSyn-Gfp*, *p<0.05; n = 6 mice (120 cells) for *pSyn-Gfp* (control), six mice (145 cells) for *pSyn-Nrg1^{ΔCt}-pSyn-Gfp*, six mice (138 cells) for *pSyn-Nrg3^{ΔCt}-pSyn-Gfp*. (**E**) Confocal images (top three panels) and binary images (bottom panel) illustrating presynaptic VGlut1+ puncta (blue) in GFP+ axons (green) of pyramidal cells located in close apposition to PSD95+ clusters (red) in PV+ interneurons (gray) in C-terminal domain deletion experiments. (**F**) Quantification of the density of VGlut1+/PSD95+/GFP+ synapses contacting PV+ interneurons in gain-of-function truncation experiments. One-way ANOVA: *F* = 19.500, p<0.001. Tukey's range test for post hoc comparison between control and experimental groups: for *pSyn-Nrg1^{ΔCt}-pSyn-Gfp*, ***p<0.001, for *pSyn-Nrg3^{ΔCt}-pSyn-Gfp*, ***p<0.001; n = 9 mice (167 cells) for *pSyn-Gfp* (control), 5 mice (107 cells) for *pSyn-Nrg1^{ΔCt}-pSyn-Gfp*, 5 mice (111 cells) for *pSyn-Nrg3^{ΔCt}-pSyn-Gfp*. Scale bars, 50 μm (**A**) and 20 μm (high magnification), and 1 μm (**C, E**). Data from the distributions of ROIs are shown as box plots, and the adjacent data points and lines represent the averages per animal and averaged mean per group, respectively. Data in synaptic quantifications represent mean ± s.e.m. The averages per animal and electroporation condition are represented in bar graphs, and the distributions of values per cell are shown in cumulative frequency plots. Data used for quantitative analyses are available in *Figure 6—source data 1*, and *Figure 6—source data 2*. The online version of this article includes the following source data and figure supplement(s) for figure 6:

**Source data 1.** Numerical data of subcellular localization in gain-of-function experiments with C-terminal domain truncated neuregulin constructs.

*Figure 6 continued on next page*

*Figure 6 continued*

**Source data 2.** Numerical data of inhibitory and excitatory synapses in gain-of-function experiments with C-terminal domain truncated neuregulin constructs.

**Figure supplement 1.** Synaptic targeting of C-terminal domain-lacking neuregulin constructs to inhibitory postsynaptic clusters in the somatic compartment of cortical pyramidal cells.

**Figure supplement 1—source data 1.** Numerical data of synaptic targeting of C-terminal domain truncated neuregulin constructs in GFP+ electroporated pyramidal cells.

**Figure supplement 2.** Targeting of C-terminal domain-lacking neuregulin constructs to excitatory presynaptic boutons innervating PV+ interneurons.

**Figure supplement 2—source data 1.** Numerical data of synaptic targeting of C-terminal domain truncated neuregulin constructs in axon terminals innervating PV+ interneurons.

Loss of ErbB4 leads to a decrease in the number of excitatory synapses received by PV+ basket cells and in the number of inhibitory synapses made by CCK+ basket cells and chandelier cells (*Del Pino et al., 2017*; *Del Pino et al., 2013*; *Fazzari et al., 2010*; *Ting et al., 2011*; *Yang et al., 2013*), and these synaptic defects are recapitulated by the summation of those reported in this study in conditional *Nrg1* and *Nrg3* mutants: loss of Nrg1 causes a decrease in the number of inhibitory synapses made by CCK+ basket cells and chandelier cells on pyramidal cells (*Figure 1* and *Figure 1—figure supplement 4*), whereas loss of Nrg3 leads to a reduction in the number of excitatory synapses received by PV+ interneurons (*Figure 1*).

Previous in vitro studies have suggested a role for Type III Nrg1 in axonal terminals of pyramidal cells (*Chen et al., 2010*; *Huang et al., 2000*; *Vullhorst et al., 2017*). These conclusions were based on the finding that exogenous Nrg1 protein expressed in cultured neurons was present in axons. The discrepancy with our results is most likely due to an abnormal subcellular distribution when exogenous constructs are expressed in neurons cultured in vitro. Our experiments reveal that only Nrg3 is transported to axon terminals of pyramidal cells in vivo, while Nrg1 is selectively retained in the somatic compartment (*Figure 3*). This is consistent with the observation that Nrg1 is required for the development of inhibitory synapses but is dispensable for the formation of excitatory synapses onto interneurons, whereas the converse occurs for Nrg3 (*Figure 2* and *Müller et al., 2018*). Together, these results demonstrate segregated functions for Nrg1 and Nrg3 in synapse development during the orchestration of cortical circuits. In the peripheral nervous system, Type III Nrg1 mediates the clustering of nicotinic acetylcholine receptors in nerve terminals through a mechanism that involves backward signaling via the phosphatidyl-inositol 3-kinase pathway (*Hancock et al., 2008*; *Wolpowitz et al., 2000*). This presynaptic function of Nrg1 suggests that the subcellular targeting of Nrg1 in peripheral sensory neurons and cortical pyramidal cells require different sorting mechanisms. It is worth noting that the conditional *Nrg1* mutant allele used in our studies lacks the EGF domain in all *Nrg1* isoforms following Cre-mediated recombination. Nevertheless, since overexpression of Type III *Nrg1* causes complementary phenotypes to those found in *Nrg1* conditional mutant mice, our results suggest that the loss of inhibitory synapses in these mice is exclusively due to the absence of Type III *Nrg1*.

A surprising finding in our study is the striking restriction of Nrg1 to the perisomatic region of pyramidal cells, as opposed to a wider somatodendritic distribution characteristic of other polarized proteins (*Farías et al., 2012*). Its specific subcellular distribution is consistent with the unique synaptic targeting of GABAergic cortical interneurons to different subcellular compartments, and in particular, the essential role of Nrg1 in perisomatic GABAergic innervation of pyramidal cells (*Figure 1* and *Figure 1—figure supplement 4*). In addition to the critical role of Nrg1 C-terminal domain in somatic targeting (*Figure 5* and *Figure 6*), there could be additional amino acid motifs in the Nrg1 sequence that coordinately participate in this peculiar sorting to confine the protein to pyramidal cell somas. A potential candidate region is the N-terminal domain of Nrg1 containing the cysteine-rich domain (CRD), which is also located in the cytoplasmic space due to the stretch of hydrophobic amino acids that conforms a second transmembrane domain to anchor the mature protein to the membrane (*Wang et al., 2001*). Interestingly, we did not observe deficits in the density of PV+ basket cell synapses contacting the soma of pyramidal cells in *Nrg1* mutant mice (*Figure 1—figure supplement 5*), which is consistent with the lack of a phenotype in these synapses in mice lacking ErbB4 from interneurons (*Del Pino et al., 2013*; *Yang et al., 2013*). Together, these data suggest that the

development of specific subsets of GABAergic synapses might be controlled by the localization of Nrg1 present in the perisomatic compartment of pyramidal cells.

Axons from pyramidal cells form abundant synaptic contacts onto cortical PV+ interneurons, a process that is regulated by Nrg3 (this study and *Fazzari et al., 2010*; *Müller et al., 2018*). Accordingly, Nrg3 was found to be specifically targeted to presynaptic boutons contacting PV+ interneurons across layers of the cerebral cortex (*Figure 3* and *Figure 3—figure supplements 4–5*), and similarly in the hippocampal CA1 circuitry (*Müller et al., 2018*). Further, our experiments showed that the specific subcellular distribution of Nrg3 depends on an intracellular mechanism whereby the cytoplasmic C-terminal domain effectively mediate the targeting of the protein to axon terminals in pyramidal cells. This raises the question of the underlying mechanisms for selective axonal sorting. One possibility is that Nrg3 C-terminal domain harbors conserved amino acid sequences that mediate a direct transport pathway to these specific presynaptic boutons through selective vesicle loading and trafficking. Since the C-terminal domain of neuregulins constitutes a large portion of the entire protein (over 350 amino acids), the identification of structural determinants potentially responsible for subcellular sorting will require further efforts. An alternative mechanism could rely on lateral diffusion and relocation to the appropriate synaptic contacts after unspecific axonal sorting, possibly mediated through binding to synaptic interacting partners (*Fu and Huang, 2010*). The co-existence of multiple mechanisms for selective axonal sorting cannot be ruled out, as it has been shown for other proteins (*Sampo et al., 2003*; *Wisco et al., 2003*; *Yap et al., 2008*).

Neuregulin binding to tyrosine kinase receptors of the ErbB family activates multiple pathways of signal transduction (*Citri and Yarden, 2006*), and previous studies have indicated that the EGF-like domain of Nrg1 is a more potent effector to trigger ErbB4 activation than Nrg3 in phosphorylation assays in vitro (*Müller et al., 2018*). Interestingly, our structure-function experiments suggest that the ErbB4-binding, EGF-like domain of both Nrg1 and Nrg3 are equally suitable to promote inhibitory and excitatory synapse formation in pyramidal cells in vivo (*Figure 4*). This observation indicates that the function of neuregulins in cortical circuit development is primarily controlled through their precise subcellular targeting, and suggests the existence of an exquisite ligand-receptor program that orchestrates the assembly of specific cortical circuits through the integration of specific inhibitory inputs and excitatory outputs in pyramidal cells. This may also provide pyramidal cells with a mechanism to independently regulate the strength of particular synapses in the adult cortex through the differential control of the expression of each neuregulin protein. Considering the strong association of neuregulin signaling with cognition and schizophrenia (*Kasnauskiene et al., 2013*; *Mei and Nave, 2014*; *Walsh et al., 2008*), our findings reinforce the idea that genetic dysfunction of interneuron-pyramidal cell connectivity might be at the core of neurodevelopmental disorders.

## Materials and methods

**Key resources table**

| Reagent type (species) or resource | Designation | Source or reference | Identifiers | Additional information |
|---|---|---|---|---|
| Antibody | Anti-Parvalbumin (polyclonal chicken) | Synaptic Systems | Cat# 195 006, RRID:AB_2619887 | IHC(1:500) |
| Antibody | Anti-CB1 (polyclonal goat) | Frontier Institute | Cat# CB1-Go-Af450, RRID:AB_2571592 | IHC(1:400) |
| Antibody | Anti-Nrg3 (polyclonal goat) | Neuromics | Cat# GT15220 | IHC(1:500) |
| Antibody | Anti-VGlut1 (polyclonal guinea pig) | Merck Millipore | Cat# AB5905, RRID:AB_2301751 | IHC(1:2000) |
| Antibody | Anti-GAD65 (monoclonal mouse, IgG2a) | Merck Millipore | Cat# MAB351R, RRID:AB_94905 | IHC(1:500) |

*Continued on next page*

*Continued*

| Reagent type (species) or resource | Designation | Source or reference | Identifiers | Additional information |
|---|---|---|---|---|
| Antibody | Anti-GAD67 (monoclonal mouse, IgG2a) | Merck Millipore | Cat# MAB5406, RRID:AB_2278725 | IHC(1:500) |
| Antibody | Anti-Gephyrin (monoclonal mouse, IgG1) | Synaptic Systems | Cat# 147 011, RRID:AB_887717 | IHC(1:500) |
| Antibody | Anti-HA (monoclonal mouse, IgG1) | BioLegend | Cat# 901502, RRID:AB_2565007 | IHC(1:500) |
| Antibody | Anti-PSD95 (monoclonal mouse, IgG2a) | NeuroMab | Cat# 70–028, RRID:AB_2307331 | IHC(1:500) |
| Antibody | Anti-Synaptotagmin-2 (monoclonal mouse, IgG2a) | ZFIN | Cat# ZDB-ATB-081002–25, RRID:AB_10013783 | IHC(1:250) |
| Antibody | Anti-DsRed (polyclonal rabbit) | Clontech, Takara Bio | Cat# 632496, RRID:AB_10013483 | IHC(1:500) |
| Antibody | Anti-HA (polyclonal rabbit) | Cell Signaling Technology | Cat# 3724, RRID:AB_1549585 | IHC(1:500) |
| Antibody | Anti-Nrg1 (polyclonal rabbit) | Abcam | Cat# ab23248, RRID:AB_2154667 | IHC(1:500) |
| Antibody | Anti-Parvalbumin (polyclonal rabbit) | Swant | Cat# PV27, RRID:AB_2631173 | IHC(1:2000) |
| Antibody | Anti-pIκBα (polyclonal rabbit) | Cell Signaling Technology | Cat# 2859, RRID:AB_561111 | IHC(1:200) |
| Antibody | Anti-chicken-DyLight 405 (donkey) | Jackson Immuno Research Europe Ltd. | Cat# 703-475-155, RRID:AB_2340373 | IHC(1:200) |
| Antibody | Anti-guinea pig-647 (donkey) | Jackson Immuno Research Europe Ltd. | Cat# 706-605-148, RRID:AB_2340476 | IHC(1:250) |
| Antibody | Anti-goat-Alexa 647 (donkey) | Molecular Probes | Cat# A-21447, RRID:AB_2535864 | IHC(1:400) |
| Antibody | Anti-mouse-Alexa 488 (donkey) | Molecular Probes | Cat# A-21202, RRID:AB_141607 | IHC(1:200) |
| Antibody | Anti-rabbit-Alexa 647 (donkey) | Molecular Probes | Cat# A-31573, RRID:AB_2536183 | IHC(1:500) |
| Antibody | Anti-rabbit-Cy3 (donkey) | Jackson Immuno Research Europe Ltd. | Cat# 711-165-152, RRID:AB_2307443 | IHC(1:500) |
| Antibody | Anti-chicken-Alexa 488 (goat) | Molecular Probes | Cat# A-11039, RRID:AB_2534096 | IHC(1:600) |
| Antibody | Anti-mouse IgG1-Alexa 488 (goat) | Molecular Probes | Cat# A-21121, RRID:AB_2535764 | IHC(1:500) |

*Continued on next page*

*Continued*

| Reagent type (species) or resource | Designation | Source or reference | Identifiers | Additional information |
|---|---|---|---|---|
| Antibody | Anti-mouse IgG1-Alexa 555 (goat) | Molecular Probes | Cat# A-21127, RRID:AB_2535769 | IHC(1:500) |
| Antibody | Anti-mouse IgG1-Alexa 647 (goat) | Molecular Probes | Cat# A-21240, RRID:AB_2535809 | IHC(1:500) |
| Antibody | Anti-mouse IgG2a-Alexa 647 (goat) | Molecular Probes | Cat# A-21241, RRID:AB_2535810 | IHC(1:500) |
| Antibody | Biotinylated anti-rabbit (goat) | Vector Laboratories | Cat# BA-1000, RRID:AB_2313606 | IHC(1:200) |
| Antibody | Biotinylated anti-rat (goat) | Vector Laboratories | Cat# BA-9400, RRID:AB_2336202 | IHC(1:200) |
| Antibody | Biotinylated anti-mouse (horse) | Vector Laboratories | Cat# BA-2000, RRID:AB_2313581 | IHC(1:200) |
| Antibody | Biotinylated anti-mouse IgG1 (rat) | BioLegend | Cat# 406603, RRID:AB_315062 | IHC(1:200) |
| Antibody | Streptavidin-Alexa 488 | Thermo Fisher Scientific | Cat# S11223, RRID:AB_2336881 | IHC(1:400) |
| Antibody | Streptavidin-Alexa 555 | Thermo Fisher Scientific | Cat# S32355, RRID:AB_2571525 | IHC(1:400) |
| Antibody | Streptavidin-Alexa 647 | Jackson Immuno Research Europe Ltd. | Cat# 016-600-084, RRID:AB_2341101 | IHC(1:200) |
| Antibody | Streptavidin-DyLight 405 | Jackson Immuno Research Europe Ltd. | Cat# 016-470-084, RRID:AB_2337248 | IHC(1:400) |
| Chemical compound, drug | DAPI stain | Invitrogen | Cat# D9542 | |
| Chemical compound, drug | Fast green | Roche | Cat# 06402712001 | |
| Chemical compound, drug | Paraformaldehye | Sigma-Aldrich | Cat# 441244 | |
| Chemical compound, drug | Ritrodrine hydrochloride | Sigma-Aldrich | Cat# R0758-1G | |
| Chemical compound, drug | Tamoxifen | Invitrogen | Cat# D9542 | |
| Chemical compound, drug | Triton X-100 | Sigma-Aldrich | Cat# T8787-100ML | |
| Sequence-based reagent | RNAscope Probe - Mm-Nrg1 | ACDBio | Cat# 418181 | |
| Sequence-based reagent | RNAscope Probe - Mm-Nrg3 | ACDBio | Cat# 441831 | |
| Commercial assay or kit | RNAscope Multiplex Fluorescent Assay | ACDBio | Cat# 323110 | |
| Genetic reagent (*Mus musculus*) | *Neurod6^Cre^* (*Neurod6^tm1(cre)Kan^*) | *Goebbels et al., 2006*, PMID:17146780 | | Dr Klaus Nave (MPI-EM) |
| Genetic reagent (*M. musculus*) | *Neurod6^CreERT2^* (*Neurod6^tm2.1 (cre/ERT2)Kan^*) | *Agarwal et al., 2012*, PMID:21880656 | | Dr Klaus Nave (MPI-EM) |

*Continued on next page*

*Continued*

| Reagent type (species) or resource | Designation | Source or reference | Identifiers | Additional information |
|---|---|---|---|---|
| Genetic reagent (*M. musculus*) | *Nrg1*^floxed (*Nrg1*^tm3Cbm) | *Yang et al., 2001*, PMID:11395002 | | Dr Carmen Birchmeier (MDC) |
| Genetic reagent (*M. musculus*) | *Nrg3*^floxed (*Nrg3*^tm1a(KOMP)Mbp) | *Bartolini et al., 2017*, PMID:28147272 | | |
| Genetic regent (*M. musculus*) | *RCL*^tdT or Ai9 (*Gt(ROSA) 26Sor*^tm9(CAGtdTomato)Hze / J) | *Madisen et al., 2010*, PMID:20023653 Jackson Laboratory | Stock: 007905, RRID:IMSR_JAX:007905 | |
| Strain, strain background (*M. musculus*) | C57BL/6J | Jackson Laboratory | Stock: 000664, RRID:IMSR_JAX:000664 | |
| Strain, strain background (*M. musculus*) | Crl:CD1(ICR) | Charles River Laboratories | Stock: 022, RRID:IMSR_CRL:022 | |
| Sequence-based reagent | Nex-4 | *Goebbels et al., 2006*, PMID:17146780 | PCR primers | GAGTCCTGGAATCAGTCTTTTTC |
| Sequence-based reagent | Nex-5 | *Goebbels et al., 2006*, PMID:17146780 | PCR primers | AGAATGTGGAGTAGGGTGAC |
| Sequence-based reagent | Nex-6 | *Goebbels et al., 2006*, PMID:17146780 | PCR primers | CCGCATAACCAGTGAAACAG |
| Sequence-based reagent | Exon1-S | *Agarwal et al., 2012*, PMID:21880656 | PCR primers | GAGTCCTGGAATCAGTGTTTTTC |
| Sequence-based reagent | Nex-ORF-as | *Agarwal et al., 2012*, PMID:21880656 | PCR primers | AGAATGTGGAGTAGGGTGAC |
| Sequence-based reagent | Cre-as | *Agarwal et al., 2012*, PMID:21880656 | PCR primers | CCGCATAACCAGTGAAACAG |
| Sequence-based reagent | Nco-1 | *Yang et al., 2001*, PMID:11395002 | PCR primers | TCCTTTTGTGTGTGTTCAGCACCGG |
| Sequence-based reagent | M7-As | *Yang et al., 2001*, PMID:11395002 | PCR primers | GCACCAAGTGGTTGCGATTGTTGCT |
| Sequence-based reagent | wt-F | *Bartolini et al., 2017*, PMID:28147272 | PCR primers | AGAGGGAGAATGGAAAACAATGAGC |
| Sequence-based reagent | wt-R | *Bartolini et al., 2017*, PMID:28147272 | PCR primers | AGATGCCAGTGTCTCTTGTTTAGGG |
| Sequence-based reagent | 131 | *Madisen et al., 2010*, PMID:20023653 | PCR primers | AAGGGAGCTGCAGTGGAGTA |
| Sequence-based reagent | 132 | *Madisen et al., 2010*, PMID:20023653 | PCR primers | CCGAAAATCTGTGGGAAGTC |
| Sequence-based reagent | 133 | *Madisen et al., 2010*, PMID:20023653 | PCR primers | GGCATTAAAGCAGCGTATCC |

*Continued on next page*

*Continued*

| Reagent type (species) or resource | Designation | Source or reference | Identifiers | Additional information |
|---|---|---|---|---|
| Sequence-based reagent | 134 | *Madisen et al., 2010*, PMID:20023653 | PCR primers | CTGTTCCTGTACGGCATGG |
| Recombinant DNA reagent | *pSyn-Gfp* (plasmid) | this paper | | See Materials and methods, Section 2. |
| Recombinant DNA reagent | *pSyn-HANrg1-pSyn-Gfp* (plasmid) | this paper | | See Materials and methods, Section 2. |
| Recombinant DNA reagent | *pSyn-HANrg3-pSyn-Gfp* (plasmid) | this paper | | See Materials and methods, Section 2. |
| Recombinant DNA reagent | *pSyn-HANrg1$^{Ct:Nrg3}$-pSyn-Gfp* (plasmid) | this paper | | See Materials and methods, Section 2. |
| Recombinant DNA reagent | *pSyn-HANrg3$^{Ct:Nrg1}$-pSyn-Gfp* (plasmid) | this paper | | See Materials and methods, Section 2. |
| Recombinant DNA reagent | *pSyn-HANrg1$^{EGF:Nrg3}$-pSyn-Gfp* (plasmid) | this paper | | See Materials and methods, Section 2. |
| Recombinant DNA reagent | *pSyn-HANrg3$^{EGF:Nrg1}$-pSyn-Gfp* (plasmid) | this paper | | See Materials and methods, Section 2. |
| Recombinant DNA reagent | *pSyn-HANrg1$^{\Delta Ct}$-pSyn-Gfp* (plasmid) | this paper | | See Materials and methods, Section 2. |
| Recombinant DNA reagent | *pSyn-HANrg3$^{\Delta Ct}$-pSyn-Gfp* (plasmid) | this paper | | See Materials and methods, Section 2. |
| Software, algorithm | FIJI (ImageJ) | National Institute of Health | RRID:SCR_002285 | |
| Software, algorithm | MATLAB | MathWorks | RRID:SCR_001622 | |
| Software, algorithm | LAS AF | Leica Microsystems | RRID:SCR_013673 | |
| Software, algorithm | Bioconductor | open source | RRID:SCR_006442 | |
| Software, algorithm | R Project for Statistical Computing | open source | RRID:SCR_001905 | |
| Software, algorithm | RStudio | open source | RRID:SCR_000432 | |

## Mice

The mouse lines *Neurod6$^{CreERT2}$* (*Neurod6$^{tm2.1(cre/ERT2)Kan}$*) (*Agarwal et al., 2012*), *Neurod6$^{Cre}$* (*Neurod6$^{tm1(cre)Kan}$*) (*Goebbels et al., 2006*), *Nrg1$^{floxed}$* (*Nrg1$^{tm3Cbm}$*) (*Yang et al., 2001*), *Nrg3$^{floxed}$* (*Nrg3$^{tm1a(KOMP)Mbp}$*) (*Bartolini et al., 2017*), and *RCL$^{tdT}$* (*Gt(ROSA)26Sor$^{tm9(CAG-tdTomato)Hze}$*) (*Madisen et al., 2010*) were maintained in a C57BL/6J background (Jackson Laboratories, #000664). CD-1 [Crl:CD1(ICR)] mice (Charles River, #022) were used for in utero electroporation (IUE) experiments. Animals were housed in groups of up to five littermates and maintained under standard, temperature controlled, laboratory conditions. Mice were kept on a 12:12 light/dark cycle and received

water and food ad libitum. All animal procedures were approved by the ethical committee (King's College London) and conducted in accordance with European regulations, and Home Office personal and project licenses under the UK Animals (Scientific Procedures) 1986 Act.

The following primer sequences were used for routine genotyping: $Neurod6^{CreERT2}$ (5' - GAGTCC TGGAATCAGTGTTTTTC - 3'; 5' - AGAATGTGGAGTAGGGTGAC - 3'; 5' - CCGCATAACCAG TGAAACAG - 3'), $Neurod6^{Cre}$ (5' - GAGTCCTGGAATCAGTCTTTTTC - 3'; 5' - AGAATGTGGAG TAGGGTGAC - 3'; 5' - CCGCATAACCAGTGAAACAG - 3'), $Nrg1^{floxed}$ (5' - TCCTTTTGTGTGTG TTCAGCACCGG - 3'; 5' - GCACCAAGTGGTTGCGATTGTTGCT - 3'), $Nrg3^{floxed}$ (5' - AGAGGGA- GAATGGAAAACAATGAGC - 3'; 5' - AGATGCCAGTGTCTCTTGTTTAGGG - 3'), and $RCL^{tdT}$ (5' - AAGGGAGCTGCAGTGGAGTA - 3'; 5' - CCGAAAATCTGTGGGAAGTC - 3'; 5' - GGCATTAAAG- CAGCGTATCC - 3'; 5' - CTGTTCCTGTACGGCATGG - 3').

## Generation of DNA constructs

Neuregulin constructs were generated by standard molecular biology procedures. We used the DNA sequences of the predominant isoforms for *Nrg1* (GenBank: NM_178591, Ensembl Transcript ID: ENSMUST00000073884.5) and *Nrg3* (GenBank: NM_008734, Ensembl Transcript ID: ENSMUST00000166968.8). The neuregulin sequences lacked the 5'- and 3'-untranslated regions (UTR) and were preceded by the Kozak consensus sequence. The different neuregulin inserts were cloned into an expression vector plasmid containing the synapsin promoter (*pSyn*) using the restriction enzymes NotI/EcoRI. These plasmids contained an additional *pSyn* promoter followed by a green fluorescent protein (GFP) as a reporter to label the electroporated cells (*Gascón et al., 2008*). For control experiments, we used the *pSyn-Gfp* plasmid lacking any neuregulin insert.

To identify the subcellular localization of neuregulins, constructs harbored a human influenza hemagglutinin (HA) epitope tag upstream of the EGF-like domain (exon 2). It has been previously reported that this tag insertion site in neuregulin loci does not alter their function (*Wang et al., 2001*). The HA tag was inserted between amino acids 222 (Leucine, L) and 223 (Serine, S) for Nrg1 protein, and between amino acids 277 (Histidine, H) and 278 (Threonine, T) for Nrg3 protein.

For EGF-like domain swapping experiments, the chimeric neuregulins were generated by replacing the EGF-like domain of a neuregulin member by the corresponding domain from its homologous gene. First, the EGF-like domain of Nrg1 (amino acids 223 to 286, corresponding to exons 2 and 3) was replaced by the EGF-like domain of Nrg3 (amino acids 278 to 346, corresponding to exons 2 and 3); this chimeric protein was named Nrg1$^{EGF:Nrg3}$. Second, the EGF-like domain of Nrg3 (amino acids 278 to 346) was replaced by the EGF-like domain of Nrg1 (amino acids 223 to 286); this chimeric protein was named Nrg3$^{EGF:Nrg1}$.

For C-terminal domain swapping experiments, the chimeric neuregulins were generated by replacing the C-terminal domain of a neuregulin member by the corresponding domain from its homologous gene. First, the C-terminal domain of Nrg1 (amino acids 326–700, corresponding to exons 6–9) was replaced by the C-terminal domain of Nrg3 (amino acids 384–713, corresponding to exons 6–10); this chimeric protein was named Nrg1$^{Ct:Nrg3}$. Second, the C-terminal domain of Nrg3 (amino acids 384–713) was replaced by the C-terminal domain of Nrg1 (amino acids 326–700); this chimeric protein was named Nrg3$^{Ct:Nrg1}$.

For C-terminal domain deletion experiments, the truncated neuregulins were generated by deleting the C-terminal domain of each neuregulin and inserting a stop codon downstream of the transmembrane domain of the protein. First, the C-terminal domain of Nrg1 (amino acids 329–700) was deleted and a stop codon was inserted after amino acid 328 (Lysine, K); this truncated protein was named Nrg1$^{\Delta Ct}$. Second, the C-terminal domain of Nrg3 (amino acids 389–713) was deleted and a stop codon was inserted after amino acid 388 (Lysine, K); this truncated protein was named Nrg3$^{\Delta Ct}$.

For in utero electroporation (IUE), neuregulin expression plasmids were used at a concentration of 1 µg/µl. DNA solution was mixed with Fast Green (Roche, Cat# 06402712001) and 1–2 µl of the solution was injected into the lateral ventricle of E14.5 embryos.

To estimate the similarity between homologous protein domains of Nrg1 and Nrg3, we used the online-based Basic Local Alignment Search Tool (BLAST) from NCBI (https://blast.ncbi.nlm.nih.gov/ Blast.cgi). Amino acid sequences were compared for exons 2–3 of both neuregulins—corresponding to the EGF-like domains—, and exons 6–9 and exons 6–10 of Nrg1 and Nrg3, respectively—corresponding to the C-terminal domains.

## In utero electroporation

In utero electroporation (IUE) was performed as described before (*Bartolini et al., 2017*). CD-1 [Crl: CD1(ICR)] mice (Charles River, #022) were used for all IUE experiments. Timed-pregnant females were deeply anesthetized with isoflurane (Piramal Critical Care Limited). Buprenorphine (Vetergesic, Ceva Animal Health Ltd) was administered for analgesia via subcutaneous injection, and ritodrine hydrochloride (Sigma-Aldrich, Cat# R0758) was applied to the exposed uterine horns to relax the myometrium. DNA solution was mixed with Fast Green (Roche, Cat# 06402712001) and 1–2 µl of the solution was injected into the lateral ventricle of embryos at E14.5. Forceps-shaped electrodes (CUY650P3, Nepa Gene) connected to an electroporator (NEPA21 Super Electroporator, Nepa Gene) were used to deliver five electric pulses (45 V for 50 ms, with 950 ms intervals). The electrodes were positioned to target cortical pyramidal cell progenitors in the subventricular zone.

## Tamoxifen injection

Tamoxifen (Sigma-Aldrich, Cat# 85256) was dissolved in corn oil (Sigma-Aldrich, Cat# C8267) (10 mg/ml) at 37°C with constant agitation. Tamoxifen at a dose of 1 mg/10 g of body weight was administered via intragastric injection into P0 postnatal $Neurod6^{CreERT2};RCL^{tdT};Nrg3^{floxed}$ or $Neurod6^{CreERT2};RCL^{tdT};Nrg3^{floxed}$ mouse pups to conditionally knock-out the corresponding neuregulin gene in cortical pyramidal cells during postnatal development.

## Histology and immunohistochemistry

Mice were deeply anesthetized with pentobarbital sodium (Euthatal, Merial Animal Health Ltd) by intraperitoneal injection, and transcardially perfused with sodium chloride solution (Sigma-Aldrich, Cat# S76530) followed by 4% paraformaldehyde (PFA) (Sigma-Aldrich, Cat# 441244) in phosphatase-buffered saline (PBS). Dissected brains were post-fixed for 2 hr at 4°C, cryoprotected in 30% sucrose (Sigma-Aldrich, Cat# S0389) in PBS, and cut frozen on a sliding microtome (Leica SM2010 R) at 40 µm. Free-floating brain slices were permeabilized with 0.2% Triton X-100 (Sigma-Aldrich, Cat# T8787) in PBS for 1 hr, and blocked for 2 hr in a solution containing 0.3% Triton X-100, 1% serum, and 5% bovine serum albumin (BSA) (Sigma-Aldrich, Cat# A8806). Then, brain slices were incubated overnight at 4°C with primary antibodies. For immunostaining using antibodies against Nrgs, heat-induced antigen retrieval (H-AR) was performed before the permeabilization step. For H-AR, brain slices were incubated in target retrieval buffer solution containing 0.01M sodium citrate and 10% glycerol (pH 6) at 70°C for 1 hr. After H-AR, brain slices were washed in PBS. The next day, the tissue was repeatedly rinsed in PBS and incubated with secondary antibodies for 2 hr at room temperature. When required, brain slices were counterstained with 5 µM 4',6-diamidine-2'-phenylindole dihydrochloride (DAPI) (Sigma-Aldrich, Cat# D9542) in PBS. All primary and secondary antibodies were diluted in 0.3% Triton X-100, 1% serum and 2% BSA. The following primary antibodies were used: chicken anti-parvalbumin (1:500, Synaptic Systems, #195 006), goat anti-CB1 (1:400, Frontier Institute, #CB1-Go-Af450), guinea pig anti-VGlut1 (1:2000, Merck Millipore, #AB5905), mouse IgG2a anti-GAD65 (1:500, Merck Millipore, #MAB351R), mouse IgG2a anti-GAD67 (1:5,000, Merck Millipore, #MAB5406), mouse IgG1 anti-gephyrin (1:500, Synaptic Systems, #147 011), mouse anti-HA (1:500, BioLegend, #901502), mouse anti-PSD95 (1:500, NeuroMab, #70–028), mouse IgG2a anti-Synaptotagmin-2 (1:250, ZFIN, #ZDB-ATB-081002–25), rabbit anti-DsRed (1:500, Clontech, #632496), rabbit anti-HA (1:500, Cell Signaling Technology, #3724), rabbit anti-parvalbumin (1:2000, Swant, #PV27), and rabbit anti-pIκBα (1:200, Cell Signaling Technology, #2859). The following secondary antibodies were used: donkey anti-chicken-DyLight 405 (1:200, Jackson ImmunoResearch Europe Ltd., #703-475-155), donkey anti-guinea pig-647 (1:250, Jackson ImmunoResearch Europe Ltd., #706-605-148), donkey anti-goat-Alexa 647 (1:400, Molecular Probes, #A-21447), donkey anti-mouse-Alexa 488 (1:200, Molecular Probes, A-21202), donkey anti-rabbit-Alexa 647 (1:500, Molecular Probes, #A-31573), donkey anti-rabbit-Cy3 (1:500, Jackson ImmunoResearch Europe Ltd., #711-165-152), goat anti-chicken-Alexa 488 (1:600, Molecular Probes, #A-11039), goat anti-mouse IgG1-Alexa 488 (1:500, Molecular Probes, #A-21121), goat anti-mouse IgG1-Alexa 555 (1:500, Molecular Probes, #A-21127), goat anti-mouse IgG1-Alexa 647 (1:500, Molecular Probes, #A-21240), goat anti-mouse IgG2a-Alexa 647 (1:500, Molecular Probes, #A-21241), biotinylated goat anti-rabbit (1:200, Vector Laboratories, #BA-1000), biotinylated goat anti-rat (1:200, Vector Laboratories, #BA-9400), biotinylated horse anti-mouse (1:200, Vector Laboratories, #BA-2000), biotinylated rat anti-mouse

IgG1 (1:200, BioLegend, #406603), streptavidin-Alexa 488 (1:400, Thermo Fisher Scientific, #S11223), streptavidin-Alexa 555 (1:400, Thermo Fisher Scientific, #S32355), streptavidin-Alexa 647 (1:200, Jackson ImmunoResearch Europe Ltd., #016-600-084), and streptavidin-DyLight 405 (1:400, Jackson ImmunoResearch Europe Ltd., #016-470-084).

## Single-molecule fluorescence in situ hybridization

Mice were perfused as described above, and brains were postfixed overnight in 4% PFA in PBS followed by cryoprotection in 30% sucrose-RNase free PBS. Brains were sectioned frozen on sliding microtome at 30 μm. Fluorescent in situ hybridization on brain slices was performed according to manufacturer's protocol (ACDBio, RNAscope Multiplex Fluorescent Assay v2, Cat# 323110). The following probes from the RNAscope catalogue were used in this study: Nrg1-C3 (ACDBio, Cat# 418181), and Nrg3-C1 (ACDBio, Cat# 441831).

## Image acquisition and image analysis

Images were acquired at 1024 × 1024 pixel resolution in an inverted Leica TCS-SP8 confocal microscope. Imaging for cell density and synapse density analyses was performed at 8-bit depth, and imaging for subcellular compartment analysis was performed at 12-bit depth. Tile scan images of brain slices were acquired in a ZEISS Apotome2. Samples from the same experiment were imaged and analyzed in parallel, using the same laser power, photomultiplier gain and detection filter settings.

For subcellular compartment analysis, images were acquired with 20X/0.50 (Magnification/ Numerical Aperture) objective, and 0.75 digital zoom at 200 Hz acquisition speed. Analysis of HA-tagged neuregulin localization in somas and neuropil was performed in MatLab (MathWorks). First, single-channel images positive for GFP, corresponding to pyramidal cell bodies, were normalized to a reference image by using a histogram matching function to allow the detection of soma and neuropil across samples using intensity-based thresholding with the same parameters. Images from the same experiment were normalized to the same reference image. For somatic compartment analysis, pyramidal cell somas—that show higher intensity of GFP+ signal compared to neuropil—were masked using a low threshold. Masks of neuregulin expression in somas were generated by thresholding of single-channel images positive for HA. The GFP+ and HA+ masks were merged, and the number of GFP-masked somas containing HA+ signal was automatically quantified. Somatic neuregulin expression was represented as the percentage of HA/GFP double-positive somas per region of interest (ROI). For neuropil compartment analysis, the neuropil of pyramidal cells was masked from the single-channel images positive for GFP using a high threshold and an additional subtraction of the area of the soma identified with a low threshold. The image thresholding method 'IsoData' was used to detect and generate masks of HA expression in the neuropil. After merging the GFP+ mask and the HA+ mask, the quantification of the percentage of HA+/GFP+ colocalization was used to estimate neuregulin expression in the neuropil of a given ROI. Imaging was performed in 3–4 slices of the somatosensory cortex, and 4–10 ROIs were quantified and averaged per animal.

For cell density analysis, images were acquired with 10X/0.30 or 20X/0.50 objectives, and 0.75 digital zoom at 200 Hz acquisition speed. Analysis of PV+ interneuron density was performed in FIJI (ImageJ) software. The number of PV+ cells was manually counted across layers in the somatosensory cortex. This number was then divided by area ($\mu m^2$) of cortex to estimate the density of cells. A minimum of six brain slices were quantified and averaged per animal. For quantification of tdTomato- or GFP-labeled pyramidal cells, the cell density analyses were performed in upper layers of the somatosensory cortex of 3–4 slices that contain the ROIs used in synaptic analyses.

For synapse density analysis, images were acquired with 100X/1.44 objective and 2.2 digital zoom at 200 Hz acquisition speed. To estimate the relative position of each neuron within the L2/3 of the cortex, we then took images of the same cells with 40X/1.40 objective and 0.75 digital zoom to measure the depth from the border between L1 and L2. Analysis of bouton/synapse densities was performed using a custom macro in FIJI (ImageJ) software, as described previously (*Favuzzi et al., 2017*). Processing of surface and synaptic single-channel images included background subtraction, Gaussian blurring, smoothing, and contrast enhancement. For quantification of somatic synaptic contacts, the PV+ or tdTomato (tdT)+ soma was detected based on intensity levels and automatically drawn to create a mask representing the surface of the cell body and to measure its perimeter.

Similarly, for quantification of axo-axonic synaptic contacts, a mask of the AIS was created from pIκBα+ structure and its length measured. For presynaptic boutons and postsynaptic clusters, a threshold of intensity was used to automatically detect putative synaptic puncta while excluding any background. The thresholds for the different synaptic markers were unbiasedly selected in a set of random images prior to quantification, and the same threshold was applied to all images from the same experiment. The 'Analyze Particles' (circularity 0.00–1.00) and 'Watershed' tools were applied to the synaptic channels, and a mask was generated. The minimum sizes for particles were defined as follows: 0.06 for GAD67+, GAD65+ and CB1R+ boutons; and 0.05 for Syt2+ and VGlut1+ boutons and Geph+ and PSD95+ clusters. Finally, a merged image of the surface and synaptic masks was created to automatically quantify the number of contacts opposed to the soma or AIS structures. The criterion to identify presynaptic boutons (GAD67+, GAD65+, CB1R+, Syt2+, or VGlut1+) contacting the surface border of a soma or AIS was that $\geq 0.04$ $\mu m^2$ of the puncta area in the synaptic mask was colocalizing with the mask of the soma or AIS. The criterion to identify postsynaptic clusters (Geph+, or PSD95+) contained inside a defined soma was that $\geq 0.04$ $\mu m^2$ of the puncta area in the synaptic mask was colocalizing with the mask of the soma. Synapses (Syt2+/Geph+, or VGlut1+/PSD95+) were identified when a presynaptic bouton and a postsynaptic cluster were contacting each other, with a colocalization area of $\geq 0.03$ $\mu m^2$ of their corresponding masks. VGlut1+/PSD95+ synaptic contacts were considered to originate from tdT- or GFP-labeled axon terminals when $\geq 0.025$ $\mu m^2$ of their area were colocalizing with the mask of tdT+ or GFP+ processes.

For the analysis of the density of endogenous Nrg puncta, images were acquired with 100X/1.44 objective and 0.75 digital zoom at 200 Hz acquisition speed, and quantitative analyses were performed using a custom macro in FIJI (ImageJ) software. For Nrg1 puncta density, ROIs (2000 $\mu m^2$) were analyzed in both the stratum pyramidale and stratum radiatum, distinguished by the location of neuronal somas labeled with NeuN. Intensity-based threshold for Nrg1 signal was unbiasedly selected in a set of random images to detect putative puncta, and the same threshold was applied to all images from the same experiment. For Nrg3 puncta density, images were processed similarly to Nrg1 density analysis, using the same criteria to detect and analyze the number of Nrg3+ clusters per ROI (24,025 $\mu m^2$). In addition, to quantitatively estimate and distinguish between Nrg3+ clusters that are in close apposition to PV+ cells—putatively representing Nrg3-labeled presynaptic excitatory inputs onto interneurons—and Nrg3+ clusters that are not in contact with PV+ cells, we used PV immunostaining to generate masks representing the cell bodies of PV+ interneurons. ROIs with similar proportion of PV+ cell body masks across genotypes were used (average, 8561.4 $\mu m^2$).

## Quantification and statistical analysis

All statistical analyses were performed using R Project for Statistical Computing (https://www.r-project.org/). For data analysis and visualization, we used the 'ggplot2' package in RStudio (https://www.rstudio.com/). Shapiro-Wilk test was used as a normality test to compare the empirical distribution function of the data sets with a normal probability distribution. To test the null hypothesis that the difference between two independent and parametric data samples measured in control and experimental conditions has a mean value of zero we used two-tailed Student's $t$-test. Mann-Whitney $U$-test was used as the non-parametric alternative test. To analyze the differences among multiple experimental groups, we used one- or two-way analysis of variance (ANOVA) test followed by Tukey's range test as a post hoc comparison test. Kruskal-Wallis test was used as the non-parametric alternative test. Statistical significance was considered at p-values<0.05. Data are presented as mean ± SEM. Number of cells or ROIs analyzed and number of animals for each experiment are described in each figure legend. Cumulative frequencies of synaptic densities are used to show the diversity of synaptic densities across all cells analyzed, further supporting the observed significant alterations in synaptic densities by depicting the general change of the entire pool of cells per experimental condition.

## Acknowledgements

We thank I Andrew and L Doglio for excellent technical assistance and E F Maraver for image analysis; C Birchmeier (Max-Delbrueck-Centrum, Berlin) for conditional *Nrg1* mice, and K Nave (Max Planck Institute of Experimental Medicine, Göttingen) for *Nex-Cre* and *Nex-Cre*ᴱᴿᵀ² mice. Supported by grants from the Medical Research Council (MRC Programme Grant, MR/S010785/1) and

European Union's Horizon 2020 research and innovation programme under grant agreement (AIMS-2-TRIALS, 777394) to BR and OM, and Fondation Roger de Spoelberch to OM. DE-A was supported by a 'la Caixa' Foundation Graduate Fellowship.

## Additional information

### Funding

| Funder | Grant reference number | Author |
|---|---|---|
| Medical Research Council | MR/S010785/1 | Oscar Marín<br>Beatriz Rico |
| Fondation Roger de Spoelberch | | Oscar Marín |
| "la Caixa" Foundation | | David Exposito-Alonso |
| European Union's Horizon 2020 research and innovation programme | AIMS-2-TRIALS, 777394 | Oscar Marín<br>Beatriz Rico |

The funders had no role in study design, data collection and interpretation, or the decision to submit the work for publication.

### Author contributions

David Exposito-Alonso, Conceptualization, Data curation, Software, Formal analysis, Validation, Investigation, Visualization, Methodology, Writing - original draft, Writing - review and editing, DE-A performed the majority of experiments described in this manuscript; Catarina Osório, Investigation, Methodology, CO performed initial experiments establishing the differential function of Neuregulins; Clémence Bernard, Investigation, Methodology; Sandra Pascual-García, Investigation, SP performed early pilot experiments with Nrg3; Isabel del Pino, Investigation, ID performed early pilot experiments with Nrg3; Oscar Marín, Beatriz Rico, Conceptualization, Resources, Supervision, Funding acquisition, Validation, Investigation, Visualization, Writing - original draft, Project administration, Writing - review and editing

### Author ORCIDs

David Exposito-Alonso (iD) https://orcid.org/0000-0002-4950-2744
Catarina Osório (iD) http://orcid.org/0000-0002-5228-0599
Clémence Bernard (iD) https://orcid.org/0000-0001-5305-7738
Sandra Pascual-García (iD) https://orcid.org/0000-0002-0536-1185
Isabel del Pino (iD) https://orcid.org/0000-0002-8672-926X
Oscar Marín (iD) https://orcid.org/0000-0001-6264-7027
Beatriz Rico (iD) https://orcid.org/0000-0002-0581-851X

### Ethics

Animal experimentation: This study was performed in strict accordance with the recommendations in the Guide for the Care and Use of Laboratory Animals in accordance with European regulations, and Home Office personal and project licenses (PPL 0808-2004-2019, PPL PD025E9BC-2019-2024) under the UK Animals (Scientific Procedures) 1986 Act. The experiments performed in this study, have been designed to follow the 3R's rules whenever possible.

### Decision letter and Author response

Decision letter https://doi.org/10.7554/eLife.57000.sa1
Author response https://doi.org/10.7554/eLife.57000.sa2

## Additional files

### Supplementary files
• Transparent reporting form

### Data availability
All data generated or analysed during this study are included in the manuscript and supporting files. Source data files have been provided for the figures and figure supplements.

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
