## [Decision Letter]

**Acceptance summary:**

All reviewers agree that your study provides important insights into a molecular program that governs the assembly of specific neuronal circuits – specifically a complementary role of Nrg1-ErbB4 and Nrg3-Erb4 in regulating inhibitory outputs and excitatory inputs of cortical interneurons, based on the differential sub-cellular trafficking of Nrg1 and Nrg3.

**Decision letter after peer review:**

Thank you for submitting your article "Subcellular sorting of neuregulins controls the assembly of excitatory-inhibitory cortical circuits" for consideration by *eLife*. Your article has been reviewed by three peer reviewers, and the evaluation has been overseen by Nils Brose as Reviewing Editor and John Huguenard as the Senior Editor. The following individual involved in review of your submission has agreed to reveal their identity: Carmen Birchmeier-Kohler (Reviewer #1).

The reviewers have discussed the reviews with one another and I have drafted this decision to help you prepare a revised submission.

As the editors have judged that your manuscript is of interest, but as described below that additional experiments are required before it is published, we would like to draw your attention to changes in our revision policy that we have made in response to COVID-19 (https://elifesciences.org/articles/57162). First, because many researchers have temporarily lost access to the labs, we will give authors as much time as they need to submit revised manuscripts. We are also offering, if you choose, to post the manuscript to bioRxiv (if it is not already there) along with this decision letter and a formal designation that the manuscript is 'in revision at *eLife*'. Please let us know if you would like to pursue this option. (If your work is more suitable for medRxiv, you will need to post the preprint yourself, as the mechanisms for us to do so are still in development.)

All three reviewers and the Reviewing Editor agreed that your article provides very interesting new insights into the mechanisms by which different types of Nrg-ErbB4 interactions differentially control the genesis of inhibitory vs. excitatory synapses in the cerebral cortex.

On the other hand, all three reviewers raised critical issues that need to be addressed before your article can be published in *eLife*. Below, I am providing a digest of the corresponding comments that we regard as essential:

Essential revisions

1) Evidence that the distribution of overexpressed HA-tagged full-length Nrgs recapitulates the distribution of endogenous Nrgs is needed to support the conclusions of the study. If immunohistochemistry is challenging, the authors could consider a molecular replacement strategy by expressing HA-tagged Nrg1 and Nrg3 in the respective Nrg conditional knockout mice.

2) Related to this issue, many data (Figure 4-6) relate to analyses of chimeric and deletion constructs of Nrg. It needs to be shown that these constructs traffic normally to the cell surface. It is possible that altered protein folding and reduced surface trafficking could result in the distribution patterns shown for the chimeric and truncated Nrg constructs, e.g. due to accumulation in endoplasmic reticulum present in somata and dendrites. Further, specific inhibitory or excitatory synaptic targeting of these constructs needs to be assessed.

3) It is unclear whether the overexpressed HA-Nrg1 that is observed in clusters on GFP+ pyramidal neurons is produced by these neurons or provided in trans. This should be clarified by quantifying perisomatic HA-Nrg1 on neighboring GFP+ and GFP- somata.

4) It appears to be assumed throughout the study that ErbB4 is located pre- or postsynaptically in the respective synapse analyzed (related to axonic vs. somatic Nrg), but this is nowhere shown. Some experiments that address the mechanism of how Nrg1/Nrg3 facilitates synapse formation should be presented. Is this dependent on a long-term (persisting) Nrg1/ErbB4 or Nrg3/ErbB4 interaction? In other words, is Nrg1-GFP located at the majority of the perisomatic synaptic sites and is ErbB4 detected there? Similarly, at how many synapses in the axonal initial segment is Nrg1-GFP/ErbB4 co-localization detected? Alternatively, a persistent Nrg1/ErbB4 or Nrg3/ErbB4 localization at the synapse might not be needed, just short-term interactions could be sufficient to elicit ErbB4 signalling.

5) In Figure 3, the localization of HA-Nrg1 to the edges of somata is visible but specific colocalization with GAD65 or especially gephyrin is not clear. Localization of HA-Nrg1 to inhibitory postsynaptic sites needs to be quantified as it is an important part of the conclusions.

6) All main conclusions depend on colocalization analyses of immunofluorescence signals and expressed markers in single confocal optical sections, with no other complementary approaches, such as electron microscopy or electrophysiology. There is a concern with this approach, as to whether individual synapses belonging to specific presynaptic and postsynaptic cells can be sufficiently resolved given the limits of standard confocal imaging. In the images shown, there is no clear correspondence between GAD65 and gephyrin, nor between VGlut1 and PSD95. This is a particular concern for excitatory synapses, as antigen retrieval methods are typically used to visualize PSD95, but do not appear to have been used here, and some of the VGlut1 puncta counted do not appear to fit entirely within the relevant tdT or GFP filled axons. Further, there is a concern that individual excitatory synapses can be resolved given their high density in cortical neuropil without using super-resolution or expansion microscopy approaches, which are now available. Currently, the best way to address this issue could be for the authors to present in an extended view figure large field images of PSD95 and VGlut1, and of GAD65 and gephyrin, showing a one-to-one correspondence of presynaptic and postsynaptic antigens and demonstrating ability to resolve all individual synapses in a field.

7) The targeting of Nrg3 to excitatory presynaptic sites on axons and presynaptic function has already been reported quite thoroughly in Muller et al., 2018. This information, as well as information on Nrg1 isoforms and their trafficking and function should be presented in the Introduction (here, there is also substantial literature, such as Vulhorst et al., 2017). In essence, much of the more general introductory background is not essential, but specific background on Nrgs needs to be added.

8) In loss-of-function experiments, a conditional Nrg1 allele was used that affects all Nrg1 isoforms, whereas in the gain-of-function experiments a type III Nrg1 cDNA is used. This should be clearly stated and discussed.

[Editors' note: further revisions were suggested prior to acceptance, as described below.]

Thank you for resubmitting your work entitled "Subcellular sorting of neuregulins controls the assembly of excitatory-inhibitory cortical circuits" for further consideration by *eLife*. Your revised article has been evaluated by John Huguenard (Senior Editor) and a Reviewing Editor.

Overall, the three reviewers agree that the revisions have improved the manuscript substantially and regard almost all critical issues resolved. However, there is one important point that remains to be addressed:

It should be tested experimentally whether ErbB4 and Nrg1/Nrg3 are co-localized at the synapses studied, and it should be quantitatively assessed what proportion of synapses contain both, the receptor and ligand. Such colocalization is implied throughout the paper, but no corresponding data are shown, nor do previously published studies adequately resolve this issue. At this point, it would be acceptable to address this issue for the wild-type scenario. Anti-ErbB4 antibodies suitable for this purpose are available.

The reviewers regard this issue to be essential because the corresponding data will have a major impact on the overall interpretation of the study.

If this information can absolutely not be provided with acceptable effort, the claim that Nrg1/Nrg3 functions are mediated by direct and persistent interactions with the ErbB4 receptor at the synapses of interest is not sufficiently supported. Accordingly, the text would have to be partially rewritten to reflect this issue – e.g. by merely proposing that ErbB4 mediates synaptic functions (e.g. based on the similarities in phenotypes of constitutive and conditional ErbB4 mutants).

---

## [Author Response]

Essential revisions1) Evidence that the distribution of overexpressed HA-tagged full-length Nrgs recapitulates the distribution of endogenous Nrgs is needed to support the conclusions of the study. If immunohistochemistry is challenging, the authors could consider a molecular replacement strategy by expressing HA-tagged Nrg1 and Nrg3 in the respective Nrg conditional knockout mice.

We thank the reviewers for this important consideration. We agree that our overexpression experiments of HA-tagged neuregulins are insufficient to provide conclusive evidence regarding the endogenous localization of Nrg1 and Nrg3 in the neocortex. Following the reviewers’ advice, we have performed heat-induced antigen retrieval-based immunohistochemistry using antibodies against Nrg1 and Nrg3 to study the endogenous localization of these proteins in the neocortex. The results of these experiments reveal the presence of endogenous Nrg1 and Nrg3 clusters in the same subcellular localisations than we previously observed the HA-tagged versions of Nrg1 and Nrg3 in pyramidal cells (Nrg1: postsynaptic to inhibitory inputs; Nrg3: presynaptic in excitatory boutons), which demonstrates that the distribution of HA-tagged Nrgs recapitulates the distribution of endogenous Nrgs.

We quantified the density of Nrg1-labelled synaptic puncta in CA1 stratum pyramidale (the natural target of somatic inhibitory boutons) and stratum radiatum (as a negative targeting control). We used *Nrg1* conditional mutant mice to demonstrate the specificity of the antibodies. The results of these experiments are shown in Figure 3—figure supplement 1.

We quantified the density of Nrg3-labelled clusters in both the CA1 region of the hippocampus and superficial layers of the cerebral cortex. We used *Nrg3* conditional mutant mice to demonstrate the specificity of the antibodies. The results of these experiments are shown in Figure 3—figure supplement 2.

2) Related to this issue, many data (Figure 4-6) relate to analyses of chimeric and deletion constructs of Nrg. It needs to be shown that these constructs traffic normally to the cell surface. It is possible that altered protein folding and reduced surface trafficking could result in the distribution patterns shown for the chimeric and truncated Nrg constructs, e.g. due to accumulation in endoplasmic reticulum present in somata and dendrites. Further, specific inhibitory or excitatory synaptic targeting of these constructs needs to be assessed.

Following the reviewers’ suggestions, we have examined the synaptic targeting of all the chimeric and truncated Nrg constructs. We have quantified the percentage of colocalisation for all constructs with inhibitory and excitatory synaptic markers. The results of these analyses have strengthened our conclusions about the specific synaptic trafficking of these constructs.

We quantified the percentage of co-localization of EGF-like domain chimeric constructs with GAD65+ puncta targeting GFP+ somas in superficial cortical layers. The results of these analyses are shown in Figure 4—figure supplement 1.

We quantified the percentage of co-localization of EGF-like domain chimeric constructs with VGluT1^+^/GFP+ puncta innervating PV+ interneurons in superficial cortical layers. The results of these analyses are shown in Figure 4— figure supplement 2.

We quantified the percentage of co-localization of C-terminal domain chimeric constructs with GAD65+ puncta targeting GFP+ somas in superficial cortical layers. The results of these analyses are shown in Figure 5—figure supplement 2.

We quantified the percentage of co-localization of C-terminal domain chimeric constructs with VGluT1^+^/GFP+ puncta innervating PV+ interneurons in superficial cortical layers. In addition, we quantified the intensity of HA+ puncta. The results of these analyses are shown in Figure 5—figure supplement 3.

We quantified the percentage of co-localization of C-terminal domain truncated constructs with GAD65+ puncta targeting GFP+ somas in superficial cortical layers. The results of these analyses are shown in Figure 6—figure supplement 1.

We quantified the percentage of co-localization of C-terminal domain truncated constructs with VGluT1^+^/GFP+ puncta innervating PV+ interneurons in superficial cortical layers. The results of these analyses are shown in Figure 6— figure supplement 2.

3) It is unclear whether the overexpressed HA-Nrg1 that is observed in clusters on GFP+ pyramidal neurons is produced by these neurons or provided in trans. This should be clarified by quantifying perisomatic HA-Nrg1 on neighboring GFP+ and GFP- somata.

The possibility that HA-Nrg1 is provided in trans is very unlikely because pyramidal cells very rarely synapse on the soma of other pyramidal cells (there are many classical studies about this, but perhaps the recent Lascone et al., Neuron 2020 provides the most comprehensive quantification). We have nevertheless quantified HA-tagged Nrg1 and HA-tagged Nrg3 clusters in electroporated GFP+ pyramidal cell somas as well as in NeuN+/GFP- neuronal somas. The results of these analyses demonstrate that HA-Nrg1 is only present postsynaptically to inhibitory terminals contacting the soma of electroporated pyramidal cells. These results also confirmed that Nrg3 is never found in that location.

We have quantified the percentage of co-localization of HA-tagged Nrg1 and HA-tagged Nrg3 with inhibitory boutons targeting GFP+ and NeuN+/GFP- neuronal somas in superficial cortical layers. The results of these analyses are shown in Figure 3—figure supplement 3.

4) It appears to be assumed throughout the study that ErbB4 is located pre- or post-synaptically in the respective synapse analyzed (related to axonic vs. somatic Nrg), but this is nowhere shown. Some experiments that address the mechanism of how Nrg1/Nrg3 facilitates synapse formation should be presented. Is this dependent on a long-term (persisting) Nrg1/ErbB4 or Nrg3/ErbB4 interaction? In other words, is Nrg1-GFP located at the majority of the perisomatic synaptic sites and is ErbB4 detected there? Similarly, at how many synapses in the axonal initial segment is Nrg1-GFP/ErbB4 co-localization detected? Alternatively, a persistent Nrg1/ErbB4 or Nrg3/ErbB4 localization at the synapse might not be needed, just short-term interactions could be sufficient to elicit ErbB4 signalling.

We agree on the interest of understanding the trans-synaptic signalling pathway mediated by neuregulins and ErbB4 receptor. We and others have previously shown the specific subcellular localisation of ErbB4 in different populations of cortical interneurons. In particular, we used electron microscopy to reveal the localisation of ErbB4 at excitatory postsynaptic densities targeting the dendrites of interneurons and at inhibitory presynaptic boutons targeting the axonal initial segment and perisomatic compartment of pyramidal cells (Fazzari et al., 2010). In the present study, we have found that the subcellular localisation of Nrg1 is specifically restricted to the perisomatic compartment of pyramidal cells, whereas the subcellular localisation of Nrg3 is greatly enriched in the axon terminals of pyramidal cells innervating PV+ interneurons. Moreover, the fact that the synaptic deficits observed in *Nrg1* and *Nrg3* conditional mutants (present study) together recapitulate those previously reported for *ErbB4* conditional mutants (our previous work: Del Pino et al., 2013; Del Pino et al., 2017), strongly suggests that presynaptic ErbB4-postsynaptic Nrg1 signalling is involved in the formation of perisomatic inhibitory synapses, whereas presynaptic Nrg3-postsynaptic ErbB4 signalling mediates the formation of excitatory synapses contacting PV+ interneurons. Although we agree that the study of the dynamic interactions between ErbB4 and Nrgs is of great interest, we think that the experiments required to asses them fall outside the scope of the current study.

The first paragraph in the Discussion summarises the genetic evidence supporting the idea that Nrg1 and Nrg3 interact with ErbB4 for the formation of the synapses described in this study.

5) In Figure 3, the localization of HA-Nrg1 to the edges of somata is visible but specific colocalization with GAD65 or especially gephyrin is not clear. Localization of HA-Nrg1 to inhibitory postsynaptic sites needs to be quantified as it is an important part of the conclusions.

We do not understand what the reviewer refers to in this comment. HA-Nrg1 is expressed post-synaptically in the pyramidal cell soma, therefore it is opposed to GAD65+ presynaptic boutons and gephyrin puncta should be embedded in the HANrg1 clusters, as shown in Figure 3 and Figure 3—figure supplement 3. To strengthen this point, we have now quantified the co-localisation of HA-tagged Nrg1 and HA-tagged Nrg3 clusters with presynaptic GAD65+ boutons targeting the somas of electroporated GFP+ pyramidal cells.

We quantified the percentage of GAD65+ boutons that are in close apposition to postsynaptic clusters expressing HA-tagged Nrg1 or HA-tagged Nrg3 in electroporated GFP+ pyramidal cells. The results of these analyses are shown in Figure 3—figure supplement 3. In this figure, we added HA staining in a colour-inverted image in low and high-magnification to more clearly illustrate the localisation of HA-tagged Nrg1 at postsynaptic locations within the pyramidal cells.

6) All main conclusions depend on colocalization analyses of immunofluorescence signals and expressed markers in single confocal optical sections, with no other complementary approaches, such as electron microscopy or electrophysiology. There is a concern with this approach, as to whether individual synapses belonging to specific presynaptic and postsynaptic cells can be sufficiently resolved given the limits of standard confocal imaging. In the images shown, there is no clear correspondence between GAD65 and gephyrin, nor between VGlut1 and PSD95. This is a particular concern for excitatory synapses, as antigen retrieval methods are typically used to visualize PSD95, but do not appear to have been used here, and some of the VGlut1 puncta counted do not appear to fit entirely within the relevant tdT or GFP filled axons. Further, there is a concern that individual excitatory synapses can be resolved given their high density in cortical neuropil without using super-resolution or expansion microscopy approaches, which are now available. Currently, the best way to address this issue could be for the authors to present in an extended view figure large field images of PSD95 and VGlut1, and of GAD65 and gephyrin, showing a one-to-one correspondence of presynaptic and postsynaptic antigens and demonstrating ability to resolve all individual synapses in a field.

We are aware of the limitations of standard confocal imaging to resolve synaptic puncta. As suggested, we now provide large field-of-view images of the distinct synaptic markers used throughout our study for both loss- and gain-of-function experiments. In addition, we have quantified the proportion of clusters co-labelled with synaptic markers that are used for co-localisation synaptic analysis. Our analyses reveal that, independently of the experimental condition, the proportion in colocalization for both inhibitory or excitatory synaptic markers is 50-70% and, more importantly, is identical for each comparison (e.g., excitatory and inhibitory synaptic terminals in conditional *Nrg1* and *Nrg3* mutants compared to their respective controls). This suggests that, even if we may not identify all synapses using confocal microscopy, we do not have artefactual biases in our analysis because we identify the same proportion of synapses in control and experimental groups. Notwithstanding these new analyses, we would like to mention that we have done this type of synaptic analyses for more than 10 years in the context of different projects, and we have repeatedly shown that our histological data is always equivalent to electrophysiological synaptic analyses performed on the same mice (Fazzari et al., 2010; Del Pino et al., 2013; Del Pino et al., 2017; Favuzzi et al., 2017; Favuzzi et al., 2019).

We have added large field-of-view images of synaptic staining in superficial cortical layers of control and *Nrg1* and *Nrg3* conditional mutant mice in Figure 1—figure supplement 3, as well as in brain slices from electroporated mice in Figure 2—figure supplement 1.

We have quantified the extent to which synaptic markers co-localize using standard confocal imaging to show that our quantitative method can reliably detect and estimate the density of synaptic clusters in brain slices of adult mice.

7) The targeting of Nrg3 to excitatory presynaptic sites on axons and presynaptic function has already been reported quite thoroughly in Muller et al., 2018. This information, as well as information on Nrg1 isoforms and their trafficking and function should be presented in the Introduction (here, there is also substantial literature, such as Vulhorst et al., 2017). In essence, much of the more general introductory background is not essential, but specific background on Nrgs needs to be added.

Following the reviewers’ suggestion, we have included these studies in the Introduction. The requested citations are now included in the Introduction.

8) In loss-of-function experiments, a conditional Nrg1 allele was used that affects all Nrg1 isoforms, whereas in the gain-of-function experiments a type III Nrg1 cDNA is used. This should be clearly stated and discussed.

We thank the reviewers for this comment. A sentence has been added in the Results and Discussion clarifying this point.

[Editors' note: further revisions were suggested prior to acceptance, as described below.]

Overall, the three reviewers agree that the revisions have improved the manuscript substantially and regard almost all critical issues resolved. However, there is one important point that remains to be addressed:It should be tested experimentally whether ErbB4 and Nrg1/Nrg3 are co-localized at the synapses studied, and it should be quantitatively assessed what proportion of synapses contain both, the receptor and ligand. Such colocalization is implied throughout the paper, but no corresponding data are shown, nor do previously published studies adequately resolve this issue. At this point, it would be acceptable to address this issue for the wild-type scenario. Anti-ErbB4 antibodies suitable for this purpose are available.The reviewers regard this issue to be essential because the corresponding data will have a major impact on the overall interpretation of the study.If this information can absolutely not be provided with acceptable effort, the claim that Nrg1/Nrg3 functions are mediated by direct and persistent interactions with the ErbB4 receptor at the synapses of interest is not sufficiently supported. Accordingly, the text would have to be partially rewritten to reflect this issue – e.g. by merely proposing that ErbB4 mediates synaptic functions (e.g. based on the similarities in phenotypes of constitutive and conditional ErbB4 mutants).

We were happy to learn that the three reviewers consider that our revisions have improved the manuscript substantially and that there is only one important point that remains to be addressed. Following your request, we have performed immunohistochemical experiments to assess, at the light microscopy level, whether ErbB4 co-localises with Nrg1 and Nrg3 at the synapses studied. We found that most ErbB4+ synaptic clusters found in PV+ interneurons also contain presynaptic Nrg3 (Figure 3—figure supplement 6), which reinforces the view that the role of Nrg3 in the formation of excitatory synapses contacting PV+ interneurons is mediated by ErbB4. We attempted to do a similar experiment for Nrg1, however we could not detect ErbB4 clusters at the inhibitory boutons at light microscopy, precluding the assessment of Nrg1-ErbB4 co-localisation. This result is not surprising since we have previously found using electron microscopy that the levels of ErbB4 at inhibitory synaptic terminals are two or three times lower than those found in postsynaptic densities being contacted by excitatory synapses (see Figure 1 and Supplementary Figures 2 and 6 in Fazzari et al., 2010). At this point, the only way to demonstrate that Nrg1 and ErbB4 co-localise at inhibitory synapses made by CCK^+^ cells on the soma and by chandelier cells on the axon initial segment of pyramidal cells is to perform double labeling immunohistochemistry for Nrg1 and ErbB4 at the electron microscopy level. We hope that the editors understand that performing this additional experiment is a major effort beyond the expected in a second revision at *eLife*. However, since we agree with the reviewers’ criticism that our experiments do not demonstrate that the function of Nrg1 and Nrg3 in synapse formation is indeed mediated by ErbB4, we have modified the text to avoid giving the impression that this is the case. In brief, we have avoided any reference to Nrg1/ErbB4 or Nrg3/ErbB4 binding or signaling in the Abstract and Results sections, and we have edited the Discussion to reflect the reasons supporting the view that ErbB4 is the most likely receptor mediating the function of Nrg1 and Nrg3 in cortical interneurons. We think the revisions introduced in the text directly address the point raised by the reviewers.